# Chewing, dentition and tooth wear in Hippopotamidae (*Hippopotamus amphibius* and *Choeropsis liberiensis*)

**Annika Avedik, Marcus Clauss** *

Clinic for Zoo Animals, Exotic Pets and Wildlife, Vetsuisse Faculty, University of Zurich, Zurich, Switzerland

* mclauss@vetclinics.uzh.ch

**Data Availability Statement:** All relevant data are within the manuscript and its Supporting Information files.

**Funding:** The author(s) received no specific funding for this work.

## Abstract

Among mammals, hippopotamids ('hippos') have been described as the species with the lowest chewing efficacy despite elaborate enamel folds on the occlusal surface or their cheek teeth, which was hypothesized to result from the lack of a grinding chewing motion. We investigated the chewing and dentition of the two extant hippo species, the common hippo (*Hippopotamus amphibius*) and the pygmy hippo (*Choeropsis liberiensis*), making (video) observations of live animals and gathering data on museum specimens (n = 86 *H. amphibius* and 26 *C. liberiensis* skulls). Hippos have a low degree of anisodonty (differences in width between maxillary and mandibular cheek teeth) and anisognathy (difference in width between the upper and the lower jaw), corresponding to a mainly orthal (up-and-down) chewing motion. The two hippo species differ slightly, but distinctively, in their anterior dental morphology and chewing mode. In both species, the canines do not completely prevent a lateral jaw movement but would, in theory, permit this movement until the mandibular canines get into contact with the maxillary protruding snout. This movement is only realized, to a small extent, in pygmy hippos, leaving distinct wear traces on their incisors and creating relatively wider wear facets on the maxillary canines. In common hippos, the interlocking upper and lower incisors prevent lateral jaw movement. Corresponding contact wear facets are evident on the medial aspect of the upper, and on the lateral aspect of the lower incisors–unless museal reconstructions mispositioned these teeth. If these facets are interpreted as an indication for a relic of a lateral jaw movement that was probably more prominent in hippo ancestors, i.e. if we assume that hippos evolved orthal chewing secondarily, several other characteristics of hippos can be explained, such as a low degree of hypsodonty (in the absence of distinct attrition due to a grinding chewing movement), a secondary loss of complexity in their enamel schmelzmuster, a secondary evolution of a wide mouth gape, a reduction in anisodonty compared to their ancestors, and the evolution of a bilaterally symmetrical ('trifoliate') enamel folding pattern on the molar occlusal surface from an ancestral bunoselenodont condition. As an underlying driving force, selection for intraspecific combat with canines and incisors, necessitating a wide gape and a rigid jaw, has been suggested.

**Competing interests:** The authors have declared that no competing interests exist.

## Introduction

Among herbivorous mammals, hippopotamuses ('hippos') show the worst chewing efficacy [1–3]. This is surprising at first glance, because hippo cheek teeth are characterized by a complex enamel folding pattern [4–6] that should serve to efficiently reduce food particle size. Because particle size is related to the speed of digestion, it has been suggested that hippos compensate for their poor chewing efficacy with particularly long digesta retention times, retaining the digesta long enough for microbial plant cell wall digestion to occur on the poorly comminuted material [7]. And because digesta retention is a function of gut volume and food intake, hippos are constrained to comparatively low food intakes [8]. The poor chewing efficacy, so to speak, constrains hippos to comparatively low food intakes [9].

The low chewing efficacy, measured by large fecal particles, has been explained by a putative lack of a transversal chewing movement caused by the anterior dentition: "*interlocking canines [. . .] prevent a grinding side-stroke. . .*" [3]. Similarly, Ryder [10] assumed a purely orthal (up-and-down) chewing stroke in hippos, without stating reasons. A closer look at the anterior dentition of hippos does not support the canine interpretation, because the arrangement of the canines does not represent the primary constraint on transversal chewing motions; rather, it is the incisor arrangement [11]. Nevertheless, hippos may serve as a contrasting example to explain why a reduction of the anterior dentition is so widespread across extinct and extant herbivorous mammalian taxa–to facilitate a higher chewing efficacy [11].

Currently, there are two extant species in the hippo family: The common hippopotamus (*Hippopotamus amphibius*) and the smaller pygmy hippo (*Choeropsis liberiensis*). They share a sister-group relationship with Cetacea [12, 13] and are classified under Cetancodonta or Whippomorpha. They belong to the order of Artiodactyls [14]. Both species have a semi-aquatic lifestyle [15] and have evolved a complex forestomach system [16–18] where plant material is digested by the aid of a microbiome [19, 20].

Common hippos mainly stay submerged underwater with only the nostrils, eyes and ears peeking out by day, and feed on land during the night. They are mainly grazers [21, 22], although some proportions of browse have been documented in some populations [23–25], and they will occasionally try to consume carrion or animals that cross the rivers they are in [26]. During a nightly feeding session, one common hippo consumes up to 60 kg of fresh grass [27, 28]. Considering the size and mass of this animal and comparing it to other herbivores, the amount, though sounding a lot, is rather low [29, 30]. However, given the restriction to feeding only at nighttime, the instantaneous intake rate may nevertheless be high. Common hippos have wide mouths which can measure up to 50 cm in width, which help to harvest grass at high instantaneous intake rates [31]. The prominent tusks and incisors are not involved in feeding, but play an important part as a defense mechanism and a weapon in the ritualized fights, which determine the relationships in the social life of hippos [32, 33]. For these fights, the lower jaw can be opened to an angle of nearly 150˚ [31].

For a long time, it was believed that the smaller pygmy hippo is a dwarf species derived from the common hippo, but more recent studies have contradicted that statement [34] and have rather shown that the pygmy hippo stems from a separate lineage dating back to the latest Miocene [35]. Pygmy hippos are generally somewhat less adapted to an aquatic lifestyle and live more closely to, or in, forests. Regarding their feeding behavior, Eltringham [29] simply suggests that the pygmy hippo "will eat whatever plants are available". Hentschel ([36], as cited in [29]) and Hendier et al. [37] investigated the plants consumed by free-ranging pygmy hippos and found that they eat grass, ferns, forbs, leaves from bushes and trees, and fallen fruits.

We aimed to describe aspects of chewing, dental anatomy, and tooth wear of both extant hippo species. In doing so, we expand the descriptions available in the literature on the lower

jaw [38] to some for the upper jaw, and we also present morphometric data for the anterior dentition. We expand on descriptions of wear traces on the incisors that indicate different constraints on chewing movements between the species [39, 40], on aspects of anisodonty (putative differences in the width of the maxillary and mandibular cheek teeth), and tooth wear stages. In particular, we suggest that wear traces on the anterior dentition indicating slight lateral jaw movements can be interpreted as relics, indicating that the loss of lateral chewing movements was secondary in hippos, which matches several other observations previously reported in the literature.

## Materials and methods

### Mastication in live animals

None of the observations or video recordings of live animals represented a deviation from regular husbandry and therefore did not necessitate animal experiment licensing. For the mastication-analysis of *H. amphibius*, we used a multitude of videos recorded at Copenhagen Zoo representing two individual hippos available via social media channels of Copenhagen hippo keeper Brian Stefanski. The videos were recorded frontally and from the side, making it possible to see the motion of the mandible and tongue during mastication as well as the movement of the incisors and the canines.

For the corresponding analyses of the pygmy hippo, we made 22 videos of two animals at the Basel Zoo, the most useful being those when animals were handfed carrots. The carrots were fed from above the animals' heads, making them lift their heads for chewing. The animals were observed and filmed from a frontal position, as well as from the side and from above, using a smartphone and a Nikon camera.

These evaluations were or a qualitative nature, describing the movement of the jaw using various anatomical reference points; they do not represent quantitative data from a large number of individuals.

### Skulls and dentition

**Sample.** We took measurements from *Hippopotamus amphibius* (n = 86) and *Choeropsis liberiensis* (n = 21) skulls from various age groups available at the Zoological Museum Zurich, the Natural History Museum Basel, the Natural History Museum St. Gallen, the Natural History Museum Berlin, the Phyletisches Museum Jena, the Stuttgart State Museum of Natural History, the Zoological Research Museum Alexander Koenig Bonn, the Senckenberg Natural History Museum Frankfurt, the Museum of Nature Hamburg, the State Museum of Natural History Karlsruhe. Seven skulls of the common hippos and four of the pygmy hippos originated from zoo animals. The origin of the remaining skulls was either unknown (common hippos n = 50, pygmy hippos n = 9) or their records confirmed that they were collected in their natural habitat (common hippos n = 29, pygmy hippos n = 8). Information regarding age and sex was generally scarce. The age of the specimen was available for one pygmy and two common hippos. In the common hippos, 6 individuals each were noted as males and 5 as females; in the pygmy hippos, six animals were listed as female and two as males.

**Qualitative observations.** We used the museum skulls to manually assess and document the possible lateral excursion of the mandible during the chewing motion. While stabilizing and slightly lifting the maxilla, we unilaterally put pressure on the mandible and deflected the lower jaw until there was opposing resistance. Additionally, the museum skulls were used to observe and describe dental anatomy, eruption sequences, and any occurring anomalies. We use the abbreviations I for incisors, C for canines, P for premolars and M for the molar teeth. Superscripts indicate maxillary and subscripts indicate mandibular teeth (e.g., $I^1$ is the first

lower and I$^1$ the first upper incisor). Permanent teeth are indicated by capital letters (I, C, P, M) and deciduous teeth by lower-case letters (i, c, p, m). The permanent dental formula of the common hippo is I 2/2, C 1/1, P 3-4/3-4, M 3/3, and I 2/1, C 1/1, P 4/4, M 3/3 for the pygmy hippo [4–6].

**Quantitative observations.** For the quantitative observations of the teeth and skulls, we used slide calipers in two different sizes and a measuring tape. All measurements were consistently taken by the first author.

*Skull and mandible.* The dimensions of the cranium (**S1 Fig in S1 File**) were gathered by measuring the distance between the *Processus palatinus ossis incisivi* (rostral margin the premaxilla) and the caudal margin of the *Os palatinum* (caudal margin of palatine), and the distance between the *Processus palatinus ossis incisivi* and the caudal margin of the *Os occipitale* (*Basis occipitale*) [terminology from 41]. The length of the mandible was measured from the caudal indentation of the *Ramus mandibulae* to the cranial margin of the *Corpus mandibulae* (**S2A Fig in S1 File**). Several distances between teeth were measured: in the upper and lower jaw, the width between the canines (**S2B Fig in S1 File**). Unfortunately, the anisognathy index as described by Fortelius [42] was not measured; it is a ratio defined as the the buccal-to-buccal distance between a certain left and right molar position (typically, the M1) of the maxilla divided by the same distance of the mandible (both distances are corrected for the width of the corresponding tooth). In the common hippo, the distance between the canine and the second incisor (or the first incisor in the case of the pygmy hippo in the lower jaw) was measured; in the common hippo, the distance between the second and the first incisor in the upper and lower jaw, and in pygmy hippos only in the upper jaw, because they only have one incisor in the mandible; the distance between the two first incisors. In some individuals, a gap was present between the third and the second premolar, which if present, was measured. If the first premolar was present, we measured the distance from P1 to the canine and to the second incisor (or to the first lower incisor in the pygmy hippos). If the first premolar was missing, we conducted the measurements starting from the second premolar.

*Incisors.* We measured the superficial tooth length on the labial and the lingual side, the tooth length between the wear facets and the mandible bone, the tooth width above the gum line and the length and width of the wear facets (**S3 Fig in S1 File**).

*Canines.* We measured the superficial tooth length of the mesial and the lingual tooth side, the tooth width above the gum line, the length and the width of the wear facet (**S4** and **S5 Figs in S1 File**).

*Premolars and molars.* All measures were performed on all teeth irrespective of wear status. We measured the tooth length in anterioposterior direction on the central axis of the tooth, as well as tooth height and maximal width (including the cingulum if present) at every cusp. The height was measured at the buccal side of the cusp, starting at the alveolar bone margin, and the width was taken from a corresponding cusp-pair (**S6A, S6B Fig in S1 File**). For all cheek teeth, we calculated the mean width, the mean height and the tooth area. The tooth area was estimated by multiplying the mean width of the tooth with the measured tooth length. We calculated the index of anisodonty introduced by Fortelius [42], by dividing the mean width of the second upper molar by the value of the second mandibular molar. The length of the tooth row was defined by the sum of the length of the molar and premolar teeth and did not include potential distances between individual teeth.

**Wear scores and age class estimates.** To estimate the age of the museum skulls, we used the age categories provided by Laws [38] for common hippos, summarized in tabulated form by Eltringham [29] (**S1 Table in S1 File**). To our knowledge, no corresponding scheme for pygmy hippos is available; therefore, we applied this scheme also to the pygmy hippo skulls.

Every skull and its teeth were ascribed to the most applicable age category. The greater the wear of the teeth, the higher the allocated age category score. The wear scores are based on the presence, morphology and connection of the dentin basins and enamel wear. We applied these stages to the mandibular and the maxillary premolar and molar teeth in both hippo species.

To quantify tooth wear of an individual animal, we composed a modified macrowear scoring table, which is based on the teeth drawings of Laws [38], with some modified or added wear stages (**S7 Fig in S1 File**).

**Data reporting and statistics.**   Mean values of measurements were calculated for the corresponding teeth of an individual, e.g. the mean length of the first lower incisor was calculated as the average of the left and right lower incisor. Because changes with age, including observations on tooth replacement, were not the major aim of the present study, they are only depicted in the S1 File, except for the development of wear with age. Several relationships were analyzed for their scaling pattern, using log-transformed values and linear regression in R [43], reporting the scaling exponent together with its 95% confidence interval (95%CI). If the 95%CI excludes 1.0, the respective scaling relationship is considered to be non-linear; in the case of length-length relationships, deviation from linearity represents a deviation from geometric scaling (or, in other words, with positive or negative allometry) [44]. Using the length of the cranium or mandible as the basis, we analyzed the scaling of the respective cheek tooth row (P2-M3) and the length of 'diastema' (the sum of the spaces between P3-P2, P2-P1 and P1-I2, or between P2-I2; using I1 if I2 not present). Similarly, we used the width of the cranium or mandible as basis and analyzed the scaling of the sum of all widths of the respective anterior dentition (I, C) and the sum of all distances between these teeth for their scaling.

## Results

### Chewing movements

**Live animals.**   We observed the mandibular movement during mastication in common and pygmy hippos. In both species symmetrical bulging of the cheeks during mastication was noted. The cheeks performed an inward motion while the mouth was opened and subsequently an outward motion while the mouth was being closed. Especially in the common hippo, the cheek motion was very prominent.

The mastication of the common hippo was evaluated on the feeding videos, by observing whether a lateral component of mandibular movement could be detected, using the various anatomical structures of the upper and lower jaw as reference points (**Fig 1**). The hippos were fed cabbage, salad and apples. There was no visible difference in the chewing motion between the different feeds. The chewing of the common hippo showed only a minimal lateral excursion of the mandible. The slight lateral motion was visible in the opposing incisor pairs (**Fig 1B**). Due to the tooth wear against the lower incisors, the originally rounded shape of the upper incisors has become lingually flattened or, especially in the case of the second incisor, even slightly arched. In the animal used in the videos, the wear facets on the lower incisors are not strongly developed. The tongue performed a forward motion in synchrony with the closure of the mouth. Through that motion, food was pressed against the occlusal area of the postcanine teeth.

During the chewing of the pygmy hippos, a slight lateral movement of the mandible was visible. The lateral movement was visually traceable focusing on the incisors and canines (**Fig 2**). In comparison to the common hippos, the incisors of the pygmy hippos show rounded tips.

**Constraints on, and traces of, lateral jaw deflection.**   We assessed the ranges of chewing motions with the museum skulls; the results of this qualitative evaluation have been presented

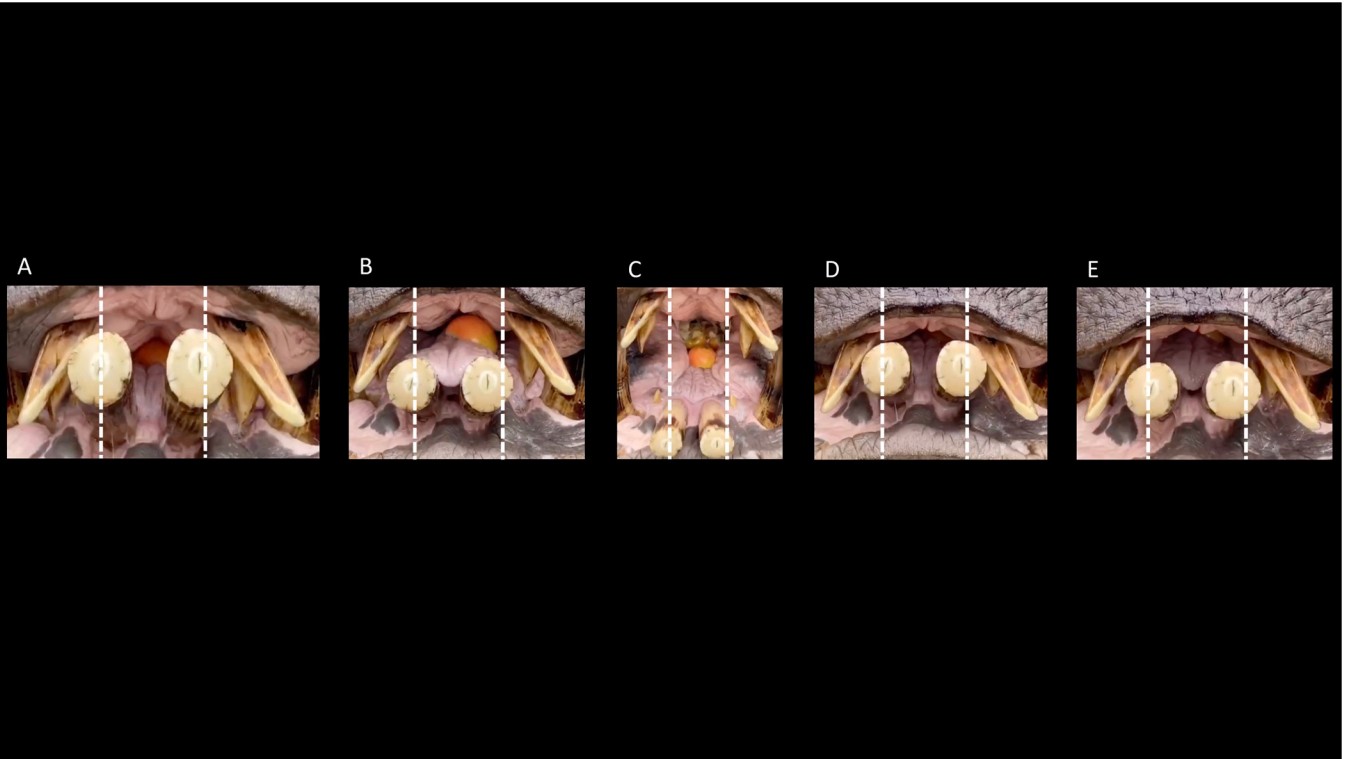

**Fig 1. The chewing and tongue motion in a *Hippopotamus amphibius*, chewing on apples.** During the opening of the mouth, the mandible shows a slight lateral deflection to the right (from the viewer's perspective to the left), which disappears with the closing of the mouth. Perpendicular interrupted lines from the lingual edge of the upper first incisor are inserted to facilitate the evaluation of a potential lateral deflection of the lower jaw. Note that the lower incisors have been artificially shortened to compensate for excessive growth, revealing a non-natural cutting surface. Video: Brian Stefanski.

in an abbreviated version in a review on how the anterior dentition can impede transversal chewing movements across mammalian herbivores [11].

In neither species did the canines interlock or impede a lateral mandibular movement; the canine wear facets allow the opposing teeth to glide both vertically and horizontally across each other. In the case of the common hippo, the width of the lower canine's wear facet does not deviate systematically from that of the upper canine (**Fig 3**), giving no particular indication of a horizontal movement. The occluding facet of their upper canines has a rather oval shape, with only a minor posterior indentation (**Fig 4A**) [35]. By contrast, in pygmy hippos the occluding facet of the upper canine has the appearance of an upside-down V shape [45] and is wider than the corresponding wear facet of the lower canine (**Fig 3**), suggesting a lateral component in the movement of these facets across each other. This upside-down V shape is caused by a deep vertical groove on the posterior aspect of the upper canine (**Fig 4B**). This groove is not present in the lower canine in either hippo species.

Two main factors constraining lateral mandibular movements were identified. The first was related to the lower canine. In both species, the lower canine was longer than the upper one, irrespective of the age of the specimen (**Fig 5**). Whereas the upper canine was, in both species, short enough to not overlap the bony mandible and hence constrain its movement, the lower canine was of a length that overlapped significantly with bony structures of the upper jaw–with the 'snout' (the *Os incisivum*).

In both hippo species, the lower canines were positioned laterally on the mandible, and the bony maxillary snout that protruded between them was sufficiently narrow to allow some

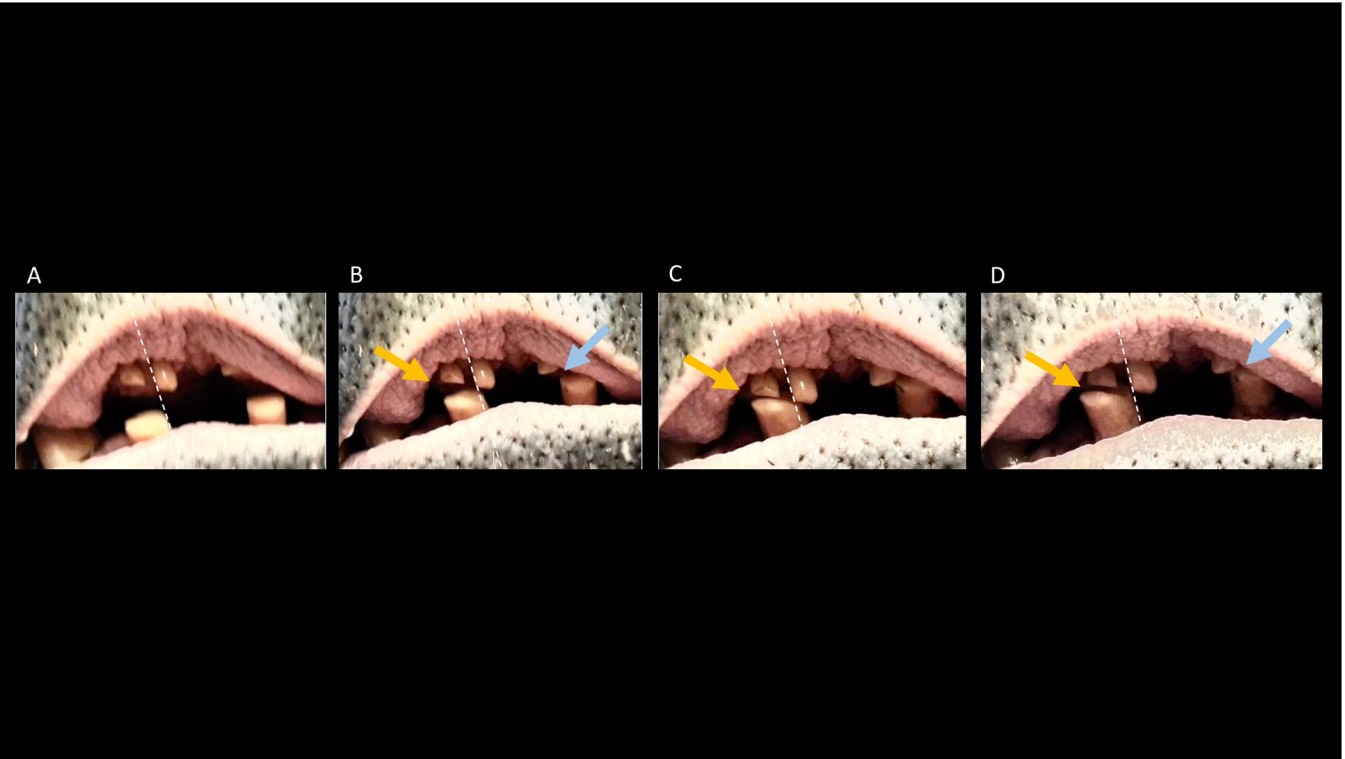

**Fig 2. The lateral incisor movement in a pygmy hippo (*Choeropsis liberiensis*), chewing on a carrot.** The dotted line along the buccal side of the right upper first incisor is inserted to facilitate the orientation of movement of the lower jaw; note that the right lower first incisor moves away from the dotted line to the animal's right, leading to the right lower incisor's buccal side being more lateral than that of the right upper second incisor's buccal side (yellow arrow), and the left lower incisor coming closer to the left upper first incisor (blue arrow).Video: Annika Avedik.

(theoretical) lateral movement of the mandible. For pygmy hippos, this lateral movement range could actually be used (**Fig 6A**), so that lateral mandibular movement was only constrained by the lower canine colliding with the snout. For the common hippo, this was not the case, as the second constraining factor applies to this species.

In the common hippos, only a minimal lateral excursion of the mandible was achieved (**Fig 6B**), due to the interlocking upper and lower incisors. While the upper incisors point vertically, the lower ones point nearly horizontally. The lower incisors are generally longer than their maxillary counterparts (**Figs 7 and 8**). The upper and lower incisors of the common hippo do not meet in occlusion. However, wear facets indicate that they come into contact during a certain lateral deflection of the mandible and that they guide the mandible into its central resting position when occlusion of the cheek teeth is reached. In the lower incisors, these wear facets are located obliquely in the central incisors and vertically in the lateral incisors [40]. The wear facets on the $I_1$ are oval-shaped and located obliquely on the lateral ('buccal') part of each tooth. They differ in size and indentation depth between individuals (**Fig 9; S3-S5; S9; S11 Figs in S1 File**). The angulation of the $I_2$ is steeper than in the first incisor. The wear facets again face the buccal direction (**Fig 9; S2B; S3; S5; S11 Figs in S1 File**). In the maxilla the wear facets of both incisors are similar in shape as in the $I_2$; they are lingually oriented, and the corresponding teeth's facets face each other (**Figs 1, 4A, 6B and 10; S1; S9; S11 Figs in S1 File**).

In the pygmy hippos, by contrast, the incisors do not create a deadlock but can glide across one another, even though mostly remaining in contact during that movement, hence leading to more apparent and pervading chewing facets (**Figs 6A and 11**). In the upper jaw, in contrast

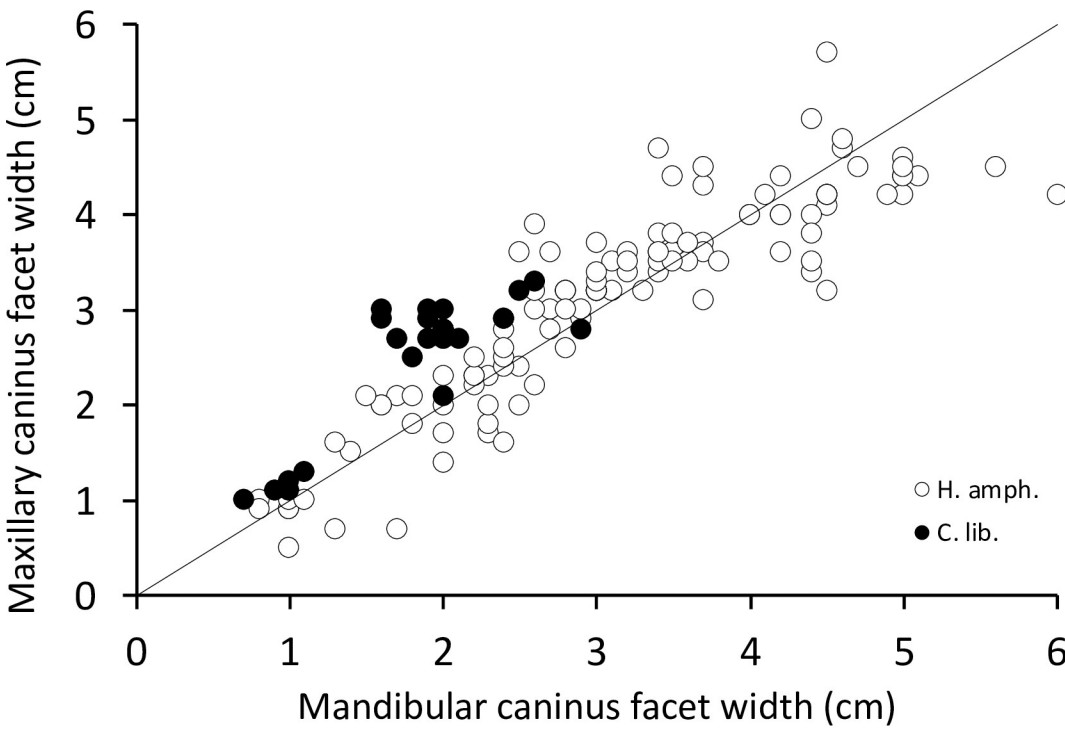

**Fig 3. Comparison of the maxillary and the mandibular canine facet width in hippos (*Hippopotamus amphibius, Choeropsis liberiensis*).** The line depicts y = x. Note the general similarity of facet widths in common and the generally larger maxillary facet width in older pygmy hippos.

to the common hippo, the whole $I^1$ is worn, often creating a facet that covers the complete maxillary incisor tip, facing downwards (**Fig 12**). On the mandibular incisors, the wear facets are horizontally oriented [40] (**Figs 6A and 11**), suggesting that the upper incisors glide across them in a lateral movement. The form and number of the facets of the lower incisors vary between individuals. The lower incisors can have a singular oval-shaped facet (**Fig 11 left**), or they can show two separate chewing facets (**Fig 11 right**). If there is only one facet, the dorsal indentation often pervades the whole width of the incisor; in the case of two facets, only one of them typically also stretches across the whole incisor width. Because the more mesial, first upper incisor is located more centrally than the second, more posteriorly positioned second upper incisor, in cases where there are two wear facets on the lower first incisor, the more mesial one will be tilted medially, and the more posterior one will be tilted laterally (**Fig 11 right**).

Because hippo incisors as well as the canines are hypselodont (i.e., they have no root) [46, 47] and, in the case of the lower incisors, have a relatively straight form and circular diameter, they are often freely movable in macerated skulls, in the sense that they can be freely rotated, or fall out. To prevent loss or movement of the material in museum collections, these teeth are often glued into their sockets. During this process, the natural position may not always be achieved. In the case of the canines and lower incisors, they might not be positioned fully in their socket, but with a more-than-normal part protruding, to create more impressive displays. Such cases can sometimes be identified by the shading on the canines or incisors (which betrays that parts of the teeth that were not exposed to the environment are now protruding from the alveolus) (**S8 Fig in S1 File**), and sometimes because the skull cannot be closed due to the obstructing, overly long canines (**S9 Fig in S1 File**). In the case of the lower incisors, in

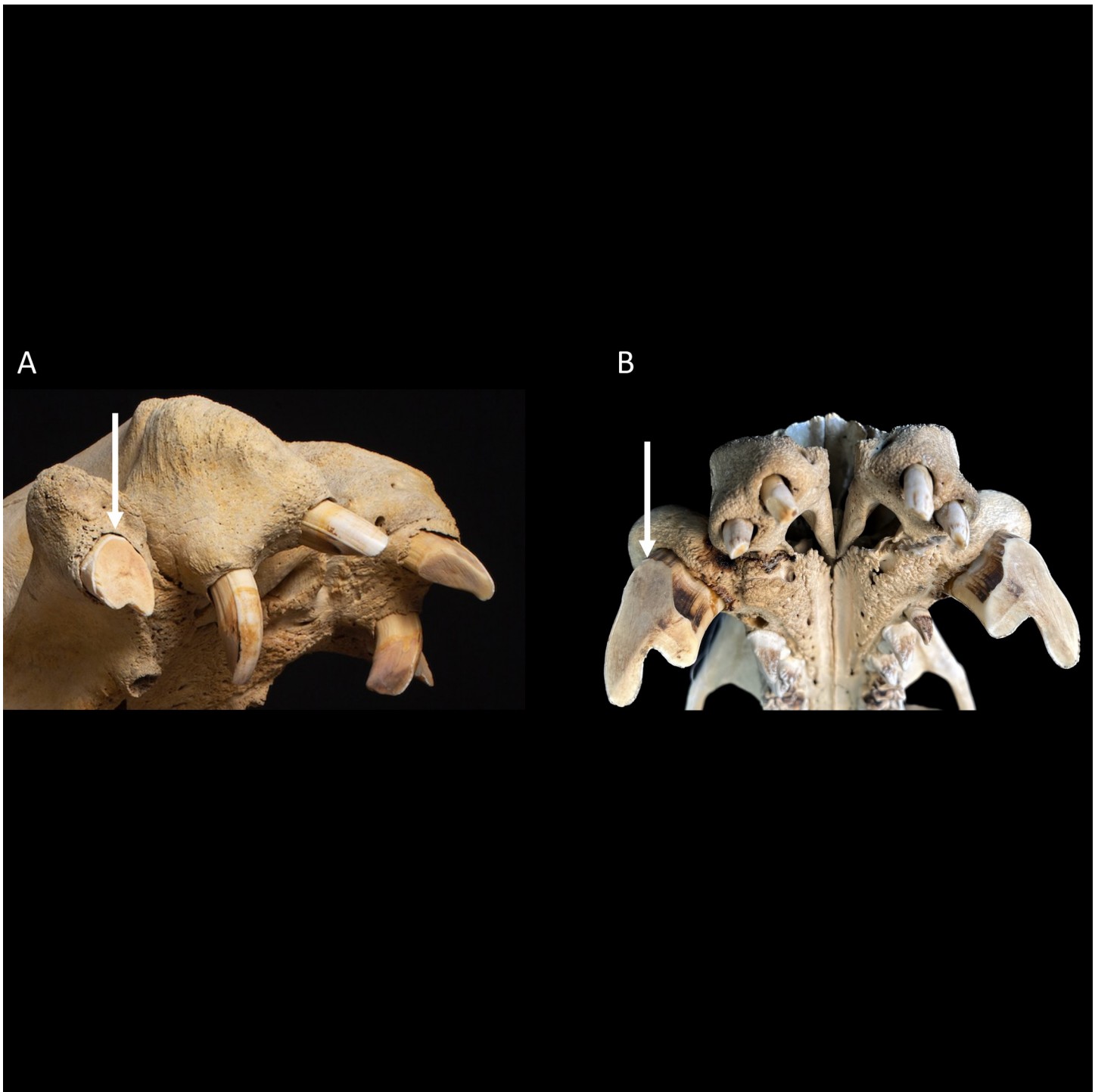

**Fig 4. (A)** Fronto-lateral view of the maxillary front teeth of a common hippo (*Hippopotamus amphibius*). The occluding tooth facet of the maxillary canine shows an oval form with only a minor posterior fossa. Note the abnormally shaped second upper incisor. Photo: Michelle Aimée Oesch. **(B)** Frontal view of the maxillary front teeth of a pygmy hippo (*Choeropsis liberiensis*). The upper canine shows a deep posterior fossa, resulting in an upside-down V-shaped tooth facet. Photo: Annika Avedik.

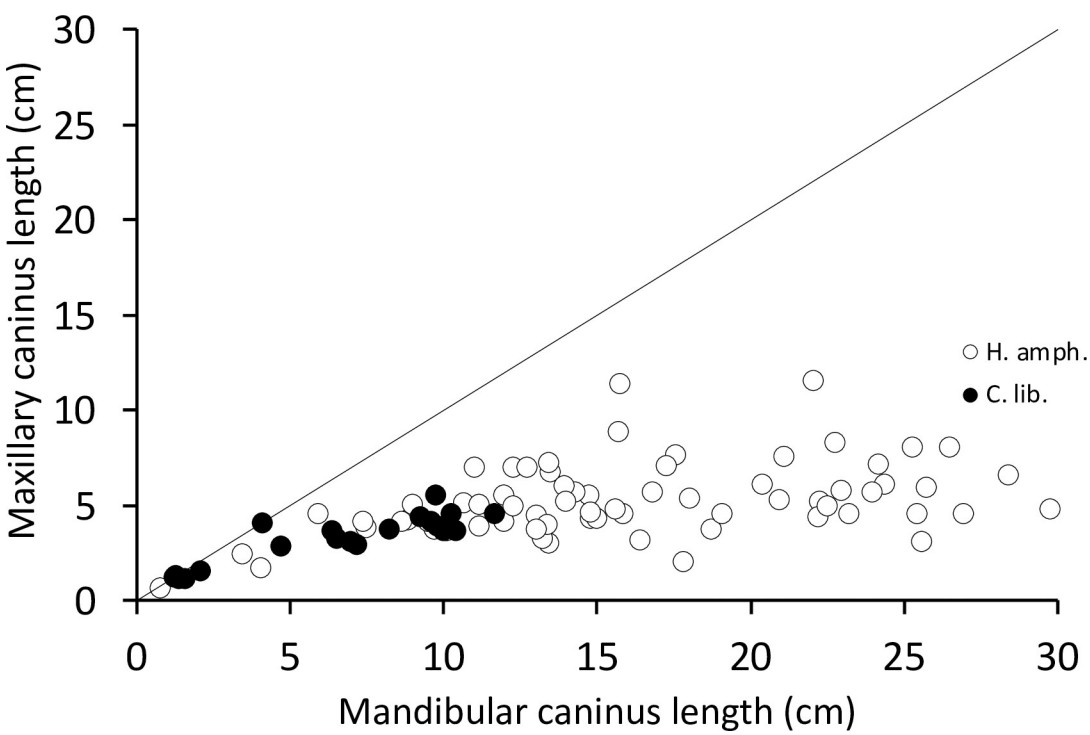

**Fig 5. Length of the maxillary vs. the mandibular canine in hippos (*Hippopotamus amphibius*, *Choeropsis liberiensis*).** The line depicts y = x. Note that the mandibular canines are systematically longer.

macerated skulls, they can be turned around their longitudinal axis in their sockets, which means that the position of facets created by the contact with the upper incisors may not necessarily be represented in their real-life position. In some cases of museum specimens, the loose front teeth have been artificially fixated into the alveoli, without consideration of the natural alignment (**S8 and S10 Figs in S1 File**). We encountered eight common hippo skulls (of n = 86) and one pygmy hippo skull (of n = 21) with mispositioned lower incisors. Thus, wear facets of the lower incisors may appear facing each other (**S8 and S10 Figs in S1 File**)–a state that cannot occur in nature, because there is no corresponding structure that can create these 'inwardly-facing' wear facets in a hippo. Therefore, the original positioning of the wear facets can only be plausibly assumed, but cannot be proven in most museum skulls. The presented interpretation of these facets can only be corroborated in live animals (**Fig 1 and S11 Fig in S1 File**).

### Anisodonty, anisognathy, tooth row length

Overall, the maxillary cheek teeth are wider than the mandibular ones in both species (**Fig 13**). In the common hippo, the anterior cusp of the $M^3$ and the cusps of the $M^2$ share the same width. In the pygmy hippo, the $M^3$, particularly its anterior cusp, is the widest tooth. (**Fig 13**). When using the second molar as the tooth of reference [42], the anisodonty is evident (**Fig 14**). We calculated the anisodonty index (ADI, introduced by [42]) by dividing the width of the $M^2$ with the width of the $M_2$. Our calculated values for the ADI for the common hippo is 1.19 ± 0.11 and for the pygmy hippo 1.25 ± 0.09. When comparing the cheek teeth for their individual cusp width, the general impression is that the dp4 and the M1 have wider posterior cusps, and the M3 a wider anterior cusp: For the dp4, this was the case in the common hippo

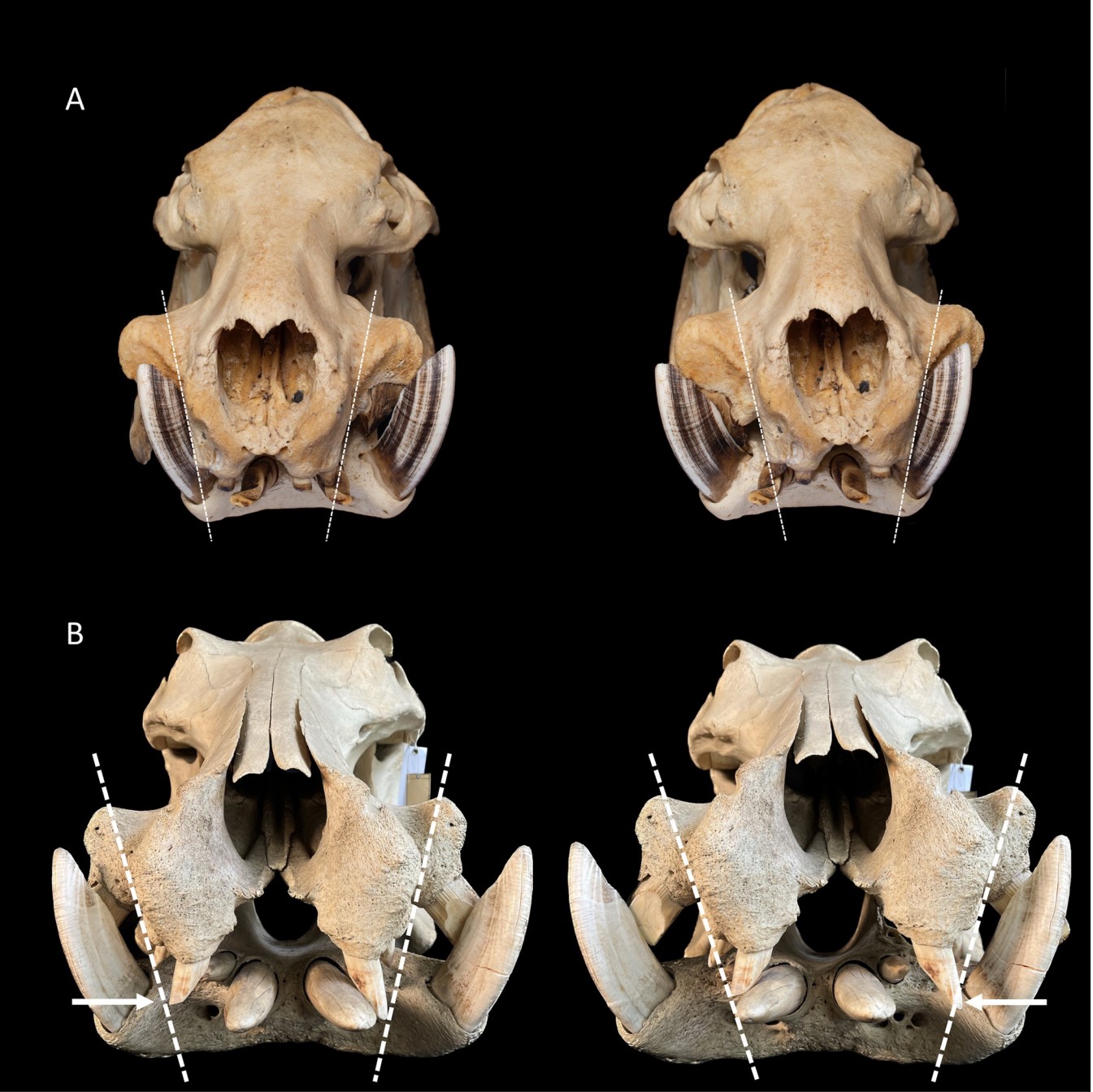

**Fig 6.** **(A)** Theoretical lateral range of motion of the lower jaw during molar occlusion in a pygmy hippo (*Choeropsis liberiensis*). Note that the range of motion is limited by mandibular canines getting in contact with the protruding maxillary snout. In live animals, soft tissue will constrain this range of motion even more. Photos: Michelle Aimée Oesch. **(B)** Theoretical lateral range of motion of the lower jaw during molar occlusion in a common hippo (*Hippopotamus amphibius*). Note that the range of motion is limited by the interlocking incisors and that the mandibular canines do not come into contact with the maxillary snout. Photos: Annika Avedik.

in 95% (maxillary) and 77% (mandibulary) of cases, and in the pygmy hippo in 89% and 84%, respectively. For the M1, this was true for the common hippo in 66% (maxillary) and 87% (mandibulary), and in the pygmy hippo in 69% and 88%, respectively. For the M3, the anterior cusp was wider in the common hippo in 92% (maxillary) and 69% (mandibulary), and in the

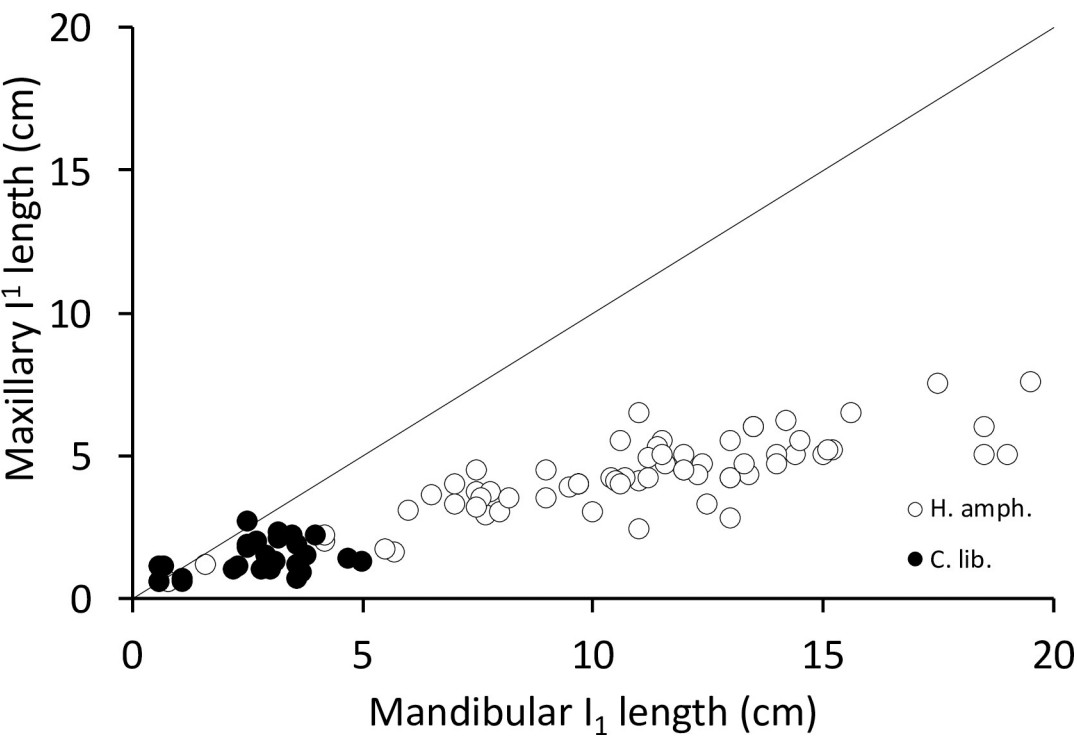

**Fig 7. Comparison of the length of the first incisor in the maxilla and mandible in hippos (*Hippopotamus amphibius*, *Choeropsis liberiensis*).** The line depicts y = x. Note that the mandibular incisors are systematically longer.

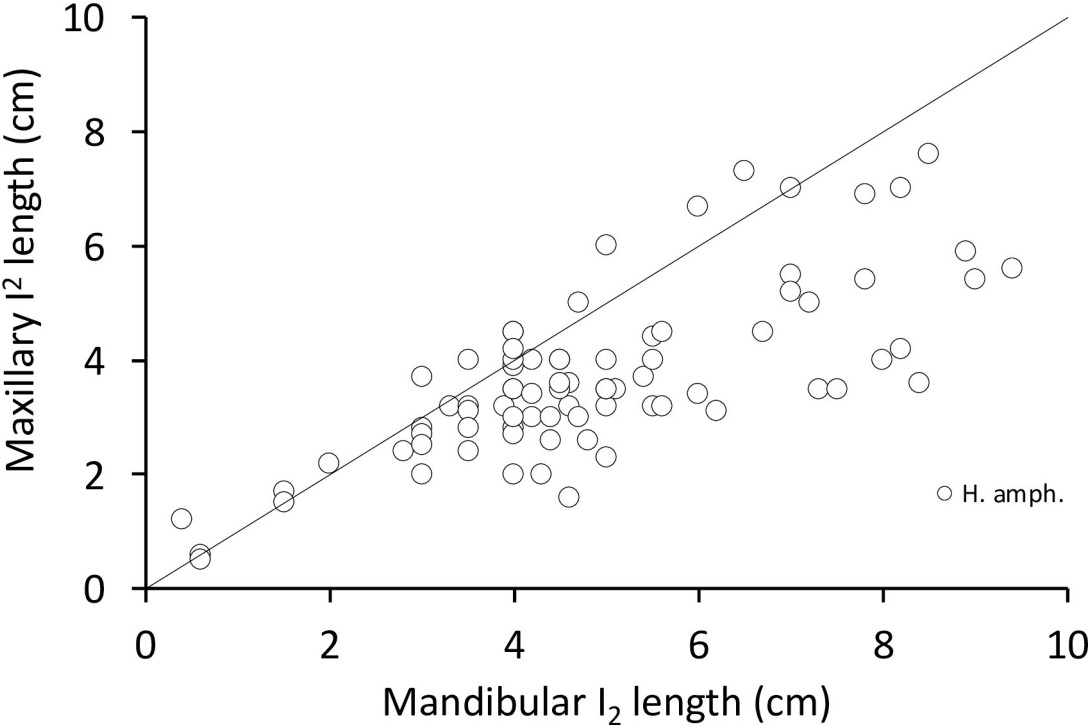

**Fig 8. Comparison of the length of the second incisor in the maxilla and mandible in *Hippopotamus amphibius*.** The line depicts y = x. Note that the mandibular incisor is systematically longer.

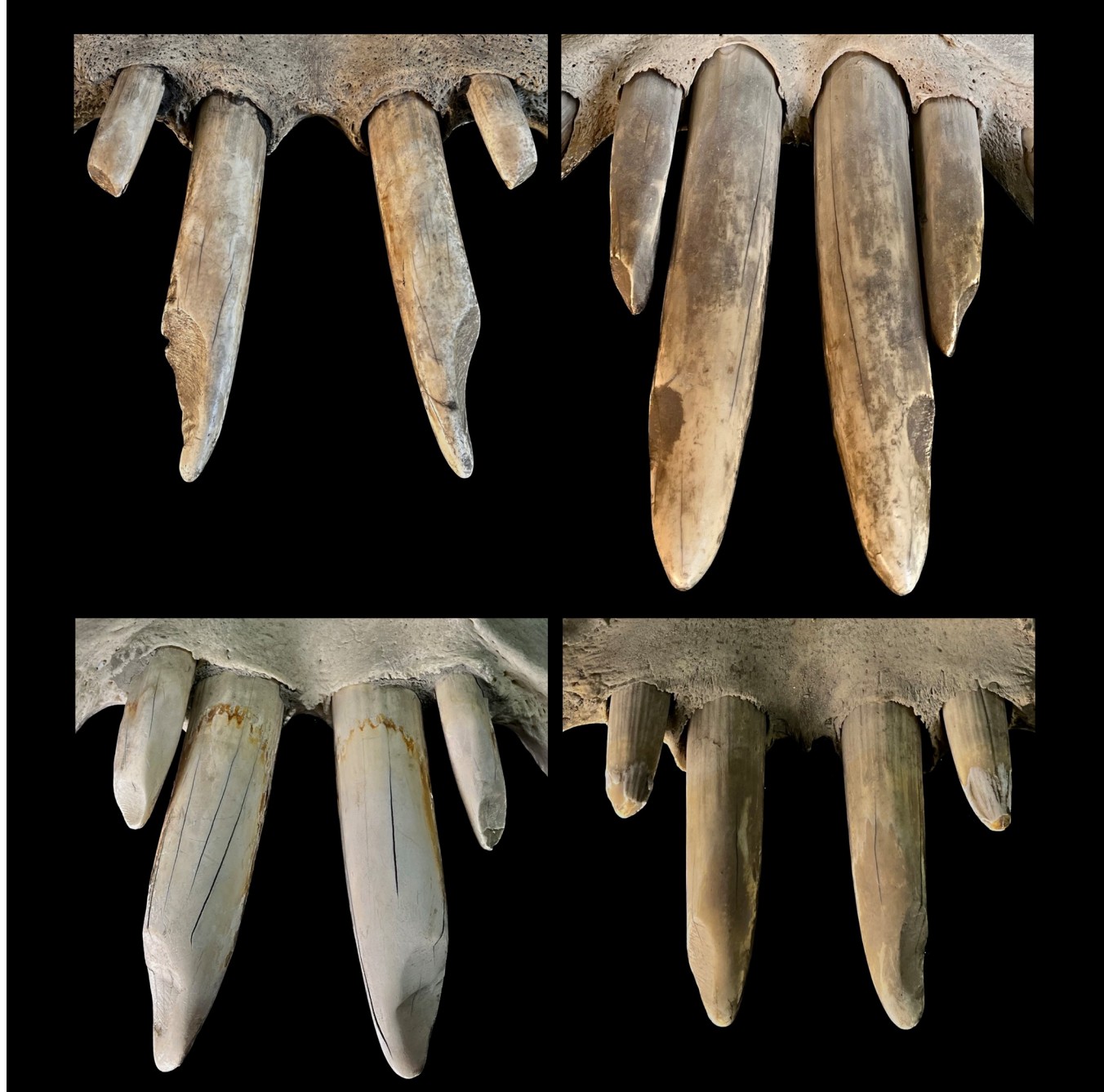

**Fig 9. Dorsoventral view of the mandibular incisors of several common hippos (*Hippopotamus amphibius*), showing the orientation of the wear traces (facets).** Photos: Annika Avedik.

pygmy hippo in 100% and 47%, respectively. The M2 had an intermediate position, with a wider anterior cusp in the common hippo in 62% (maxillary) and 17% (mandibulary), and in the pygmy hippo in 58% and 5%, respectively (**Fig 13, S12-S16 Figs in S1 File**). This most likely reflects the width of their respective osseous bases of the maxilla or mandible at the time of tooth formation.

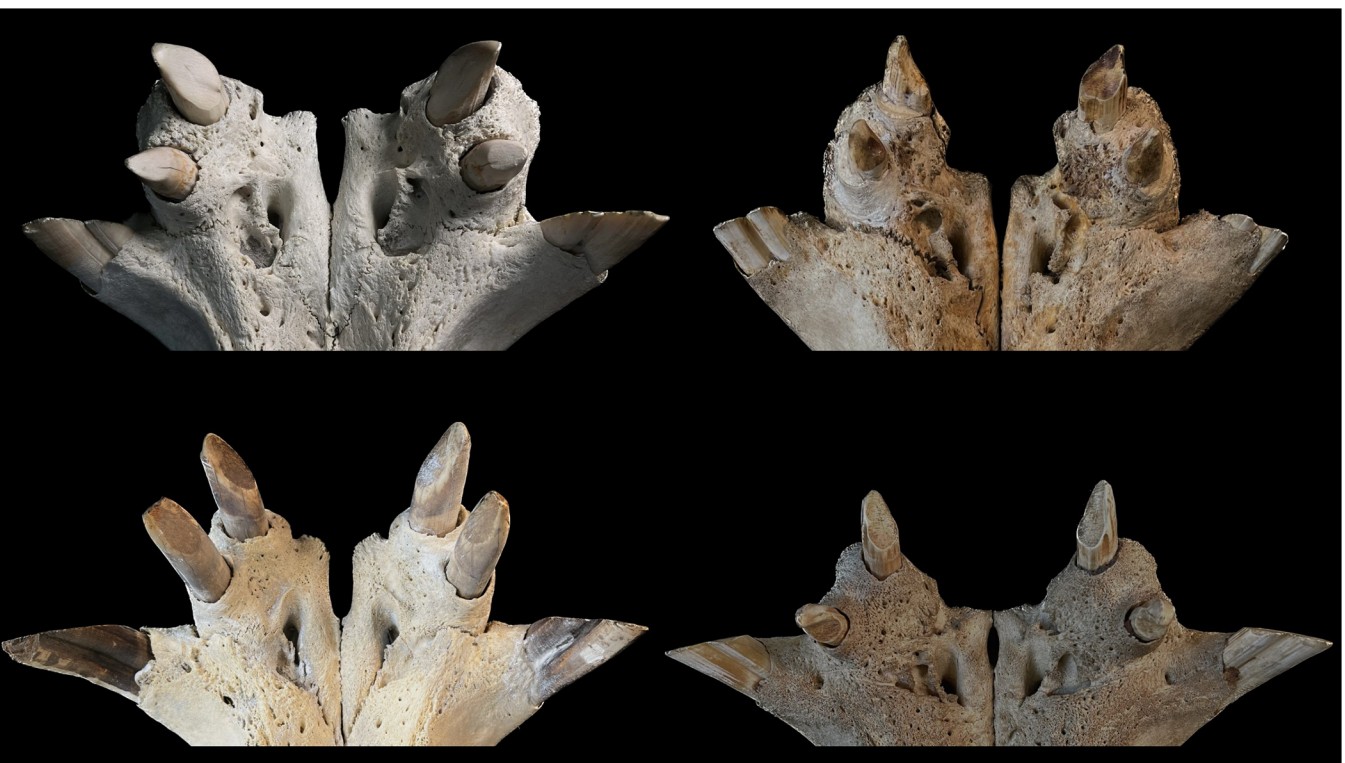

**Fig 10. Ventrodorsal view of the maxillary incisors of several common hippos (*Hippopotamus amphibius*), showing the orientation of the wear traces (facets).** Photos: Annika Avedik.

In both hippo species, the mandibular tooth row length exceeds the maxillary length, which is also (but not exclusively) linked to the additional posterior cusp of $M_3$ (**Fig 15**). The tooth length increases in the more posterior teeth (**Fig 15**). In the common hippo, all molars and also the P4 tend to be longer in the mandible (**Fig 15**, **S17 Fig in S1 File**). The upper and lower teeth are generally of similar length in the pygmy hippo, except for the $P^1$ (which is longer than the $P_1$) and the $M_3$ (which is longer than the $M^3$, due to the additional posterior cusp on the $M_3$) (**Fig 15**, **S17 Fig in S1 File**). In both hippo species, all cheek teeth are generally longer than wide, with the exception of the $P^4$, $M^1$ and $M^2$ (**S18, S19 Figs in S1 File**). When expressing the length of the cheek tooth row represented by the premolar (P2-P4) as % of the total cheek tooth row (P2-M3), the resulting values were 40.7 ±0.01 and 39.0 ±0.02% for the common hippo maxilla and mandible, and 43.6 ±0.03 and 40.5 ±0.01% for the pygmy hippo maxilla and mandible, confirming the observation that this percentage is higher in the pygmy hippo [48].

In both species, the $M^1$ and $M^2$ generally have a larger occlusal area than their mandibular counterparts; for the M3, the opposite is true in the common hippo, and the areas of the $M^3$ and $M_3$ are similar in the pygmy hippo (**S20, S21 Figs in S1 File**). When comparing the complete molar tooth row, the maxillary occlusal surface is larger in both species than the mandibular one (**Fig 16**).

When comparing the width between the lower and upper canines, the mandible was generally wider in common hippos at this site, and the difference even increased with size (and hence age) (**S22 Fig in S1 File**). In pygmy hippos, the opposite pattern appeared with the maxilla being wider at this site than the mandible in larger (and hence older) individuals (**S22 Fig in S1 File**).

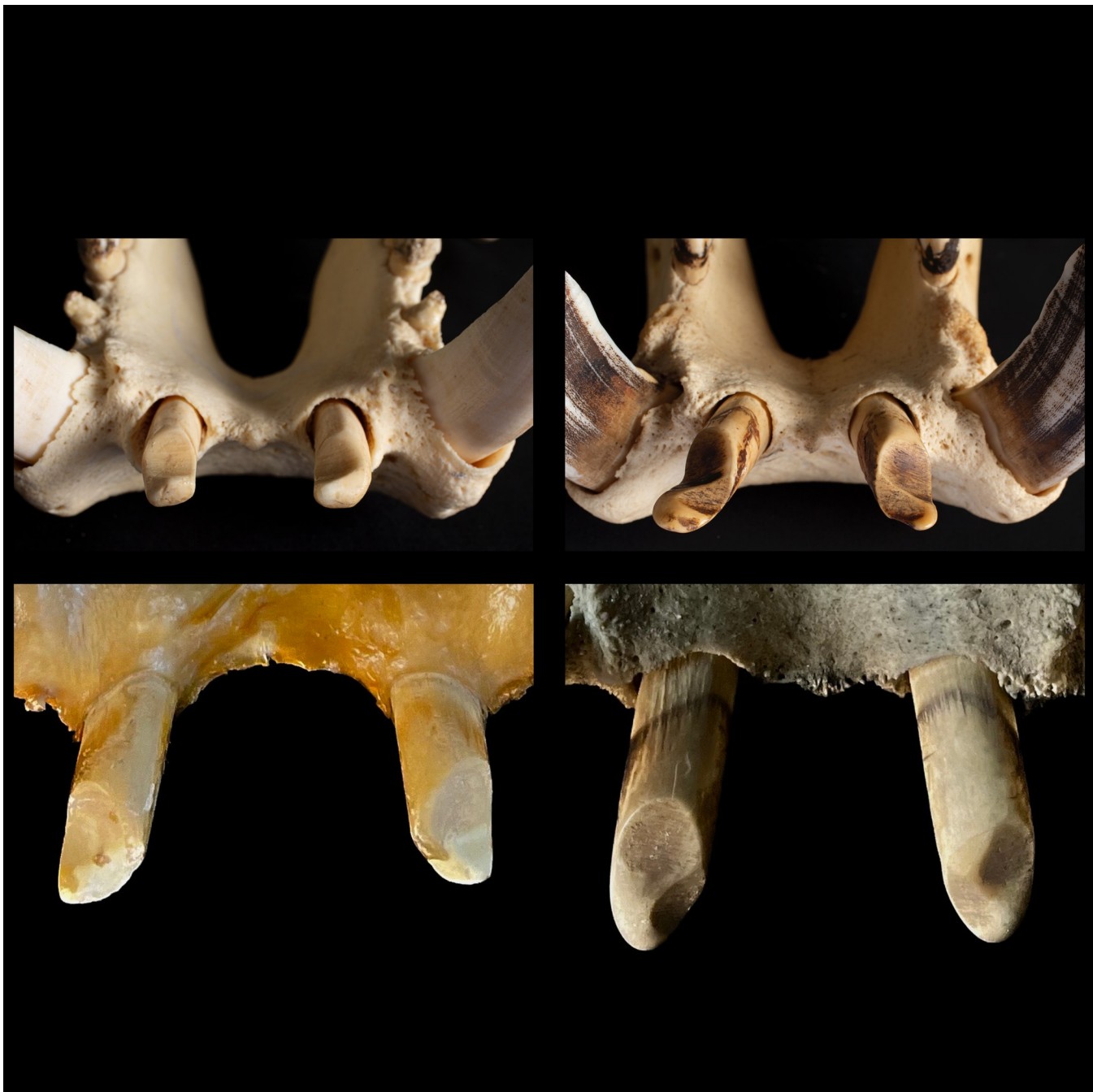

**Fig 11. Frontal view of the mandibular incisors of several pygmy hippos (*Choeropsis liberiensis*), showing the horizontal orientation of the wear traces (facets).** On the left, examples of incisors with a single wear facet; on the right, examples of incisors with two wear facets. Photos: Michelle Aimée Oesch, Annika Avedik.

Plotting the upper and lower cheek tooth row length to the length of the skull and the mandible, respectively, in adult specimens, it is evident that variations in body size (as reflected in cranium or mandible length) do not translate linearly into variations in tooth row length within common hippos (i.e. there is negative allometry between tooth row length and skull length), whereas across the two species, this relationship is close-to-linear (**Fig 17A and 17B**).

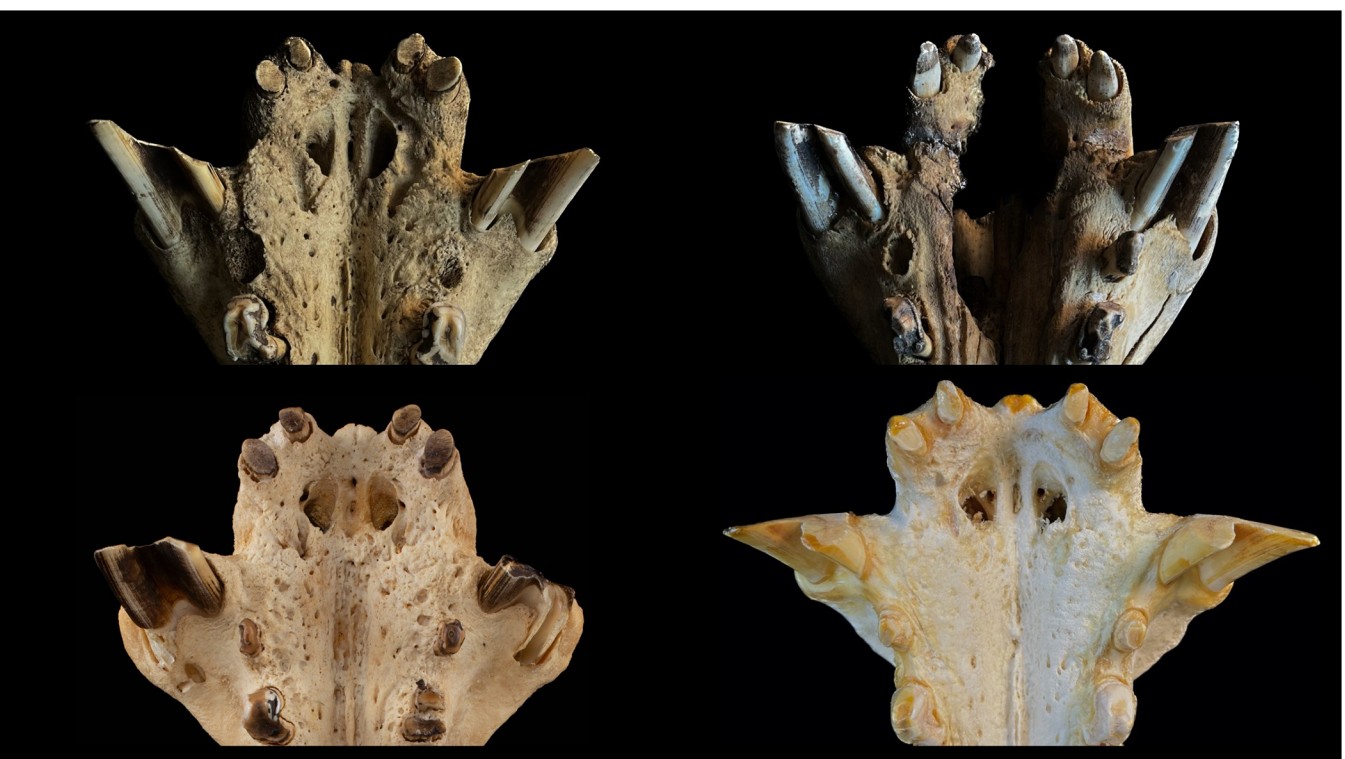

**Fig 12. Ventrodorsal view of the maxillary incisors of several pygmy hippos (*Choeropsis liberiensis*), showing the horizontal orientation of the wear traces (facets).** Photos: Michelle Aimée Oesch, Annika Avedik.

Diastema length increases with the respective skull length at a more-than-linear scaling (positive allometry) within the common hippos, but for the mandible, linear scaling is not excluded across the two species (**Fig 17C and 17D**). When comparing maxilla or mandible width to the width of anterior teeth or the space between the anterior teeth, most scaling relationships for the cranium included linearity within common hippos and across the two species, whereas for the mandible, the sum of anterior tooth width increased more-than-linearly with mandible width in common hippos and the space between the anterior teeth nearly stayed constant across body sizes; across both species, the tooth width scaling did not exclude linearity, but that of the interdental spaces was below linearity (**S27 Fig in S1 File**).

## Tooth wear

Plotting tooth height against the age classes results in the expected pattern of a more or less distinct initial increase with age (representing tooth eruption) and a subsequent decrease of tooth height (representing wear) (**Fig 18, S28, S29 Figs in S1 File**). For both species, the individual specimens identified as zoo animals mostly had relatively high teeth for the estimated age class (**Fig 18, S28, S29 Figs in S1 File**). However, one must remember that age was not measured independently, but by the visual patterns of tooth eruption and wear.

Generally, tooth wear develops at the same rate on the mandibular and the maxillary cheek teeth (**Fig 19A and 19B**). Due to their fundamentally different shape, it is of little use to compare wear stages between premolars and molars. Among the molars, there is a clear sequence of wear, with a higher wear stage for the M1 than the M2 than the M3 within an individual (**Fig 20A and 20B**).

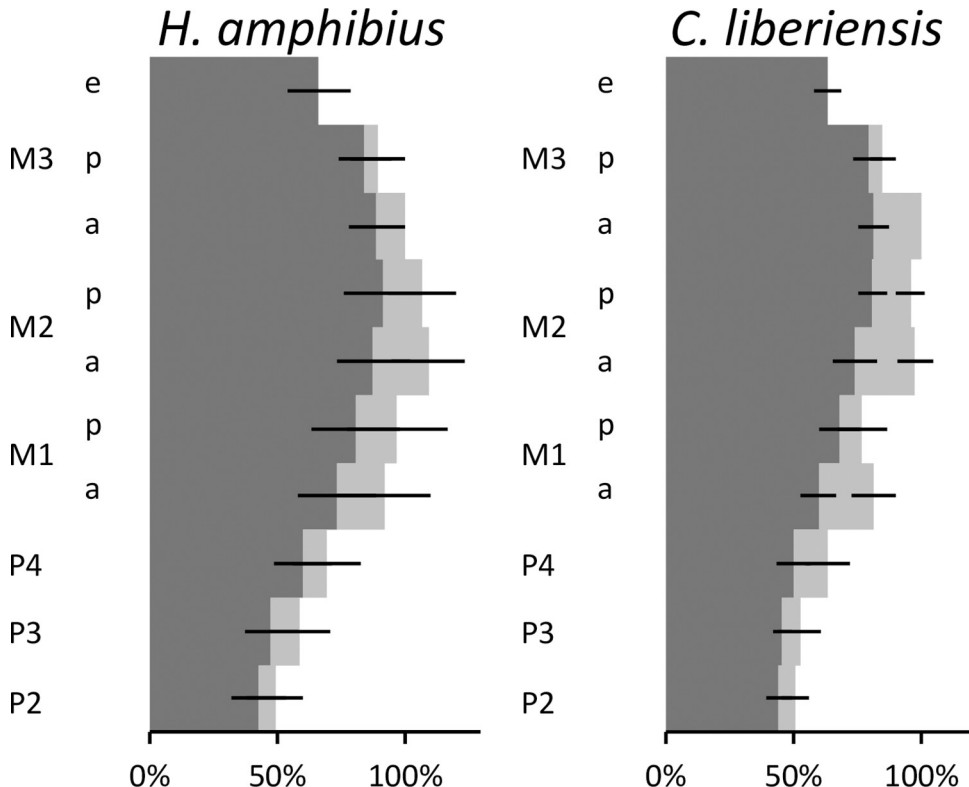

**Fig 13.** Relative width of the anterior (a) and posterior (p) cups (with e designating the additional cusp on the $M_3$) of individual teeth in relation to the width of the anterior cusp of the $M^3$ (which is set at 100%) in hippos (*Hippopotamus amphibius*, *Choeropsis liberiensis*). The maxillary tooth row is illustrated with a lighter shade, compared to the mandibular, overlying tooth row. Note that the $M^3$ only has two cusps, whereas the $M_3$ has three cusps. Data only from specimens with all teeth. Dark lines represent the standard deviation.

In the common hippo, there was no indication for a systematic difference in tooth wear between free-ranging and the two zoo animals for which all three molars were available, even though one animal had a lower wear score for the M1 than the M3 (**Fig 20A**). In the pygmy hippo, the two zoo animals appeared to have slightly less wear on the M1 and M2 compared to free-ranging animals of the same M3 wear stage (**Fig 20B**). The only remarkable pattern in the wear stage distribution was that while the M3 was passing through wear stages 3 to 6 (from the first eruption through the bone to the first wear signs), little additional wear was recorded on the corresponding M1 and M2, suggesting that the eruption occurs comparatively rapidly.

## Discussion

Our study documents well-known facts about the dental morphology of hippos with an additional morphometric dataset that may be useful in future comparative evaluations. Most specifically, we suggest that the wear traces on the incisors indicate that both species actually attempt a certain degree of lateral jaw movement that is, in common hippos, prevented by interlocking incisors. In both species, the incisors are worn not as an effect of food acquisition, but mainly due to their contact during chewing related to particle size reduction by the cheek teeth. Hippos show a distinct yet low degree of anisodonty. Thus, our study supports the concept that a lateral chewing motion was most likely ancestral to hippopotamids, yet today only occurs to a limited extent in pygmy and hardly in common hippos. In the following, we discuss

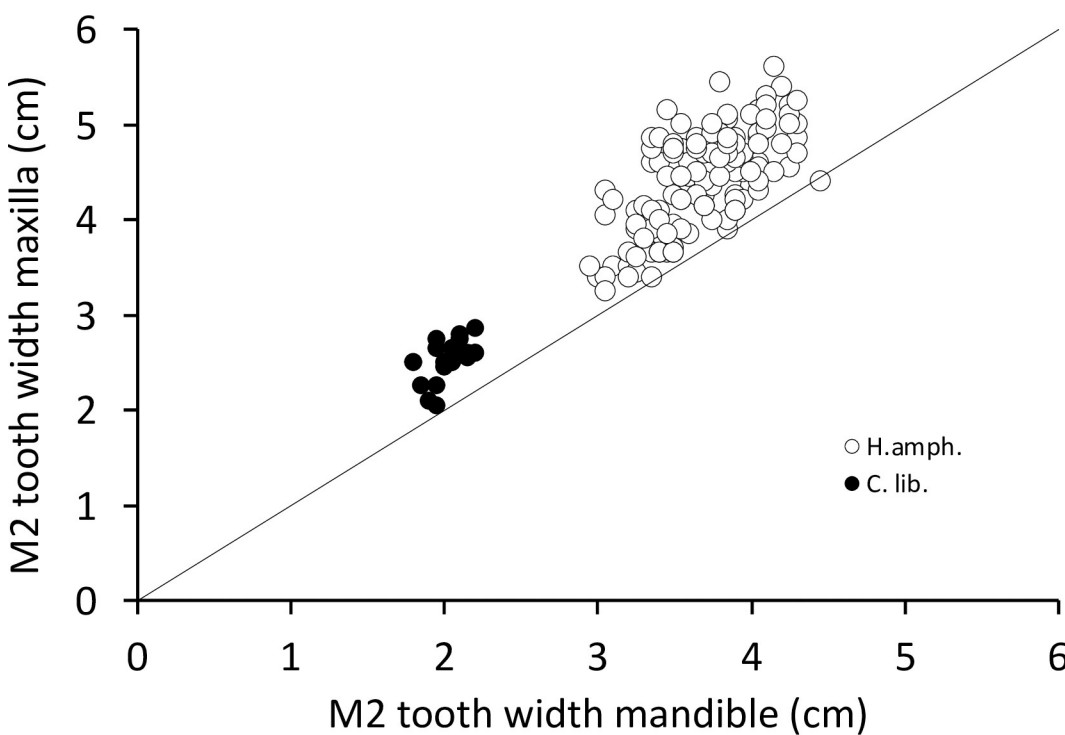

**Fig 14. Comparison of the mean width of the second molar in the mandible and maxilla in hippos (*Hippopotamus amphibius*, *Choeropsis liberiensis*).** The line depicts y = x. Note that the maxillary $M^2$ is systematically wider.

some selected features of hippo dental morphology and how the assumption of a secondary loss of a lateral chewing movement can be reconciled with other peculiarities of hippo dentition.

First, some limitations of the present study need to be mentioned. Our observations of live animals were based on a comparatively small sample of individuals, and corroboration of these observations would be welcome; nevertheless, the congruence with wear traces on museum specimens, and the consistency with which the described incisor wear facets can be observed across museum specimens (**Figs 1, 2, 6 and 9–12; S1; S2-S5; S9 Figs in S1 File**), support the interpretation. Additionally, the configuration of the incisors, including the difference in the wear facets suggesting some lateral jaw movement in pygmy and prevention of the same in common hippos, can also be gleaned from various published images of hippo skulls, although these facets are not specifically mentioned, e.g. Figs 18, 19 of [49], Figs 8–11 of [50], Fig 629 of [4], Figs 3, 4 of [46], Fig 12C of [51], Fig 2 of [52], Fig 2 of [15]. Fig 629 of [4] implies that the two different conditions of the wear facets on the lower incisors in pygmy hippos depicted in our study in **Fig 11** can actually occur within the same specimen; in that figure, a pygmy hippo's right $I_1$ displays two distinct wear facets, whereas its left $I_1$ displays only one large wear facet across its whole dorsal width–a configuration not observed in the present study. A major limitation of the present study was the limited information available on the origin and the sex of the animals. Therefore, comparisons between animals from natural habitats and zoos (**Figs 18 and 20**), and with respect to the growth of the skull and teeth between the sexes (**S23-S26 Figs in S1 File**), were not feasible with any confidence. Our anecdotal observation that dental wear does not appear to be particularly pronounced in zoo animals matches the absence of reports that zoo hippos are susceptible to excessive wear [53]–only to excessive incisor or canine growth. Finally, although some skulls of juvenile animals were included in our study

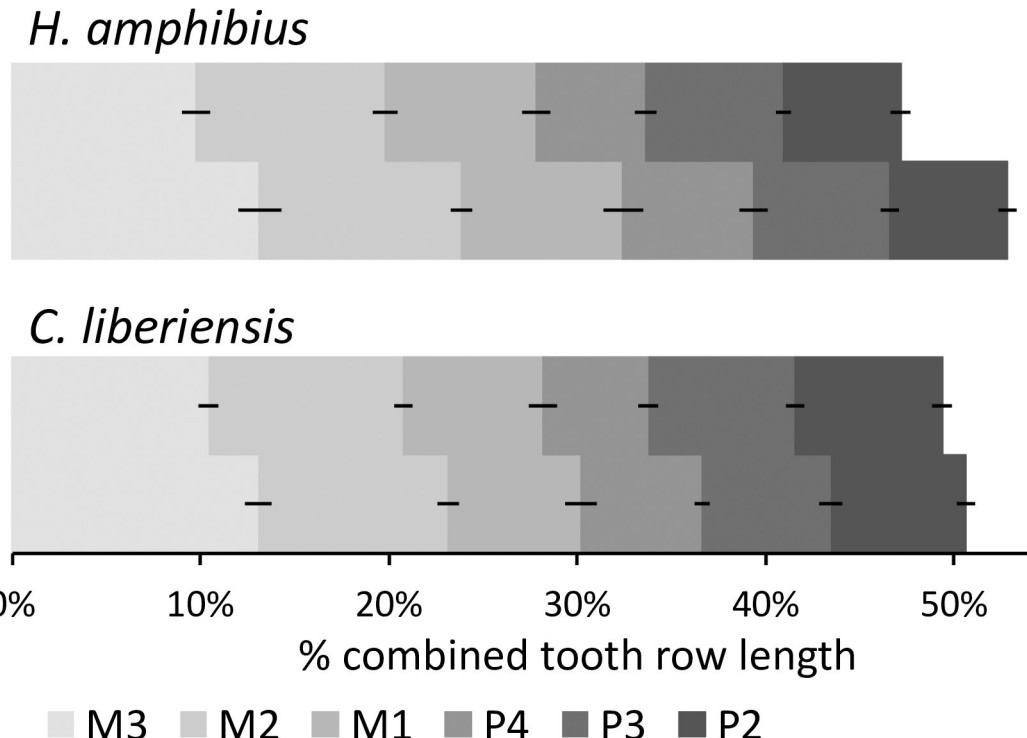

**Fig 15. Length of individual teeth of the maxillary and mandibular cheek tooth row in relation to the total length of both rows combined (100% = length of maxillary plus the mandibular cheek tooth row) in hippos (*Hippopotamus amphibius*, *Choeropsis liberiensis*).** Data only from specimens with all teeth. Dark lines represent the standard deviation.

(**S30-S38 Figs in S1 File**), we did not assess at what age the constraining effect of the canines and the incisors on transversal chewing should become pronounced. For example, the drawing of Laws [38] suggest that this might become particularly effective only at the age of 3–4 years in common hippos. Given that common hippos are weaned between one and two years of age [54], this would theoretically mean that there could be a short ontogenetic window in which digestion of solid food supported by lateral chewing could occur. Whether hippos actually use this opportunity, or whether the general arrangement of the cranial skeleton and muscles nevertheless prevents this, remains to be investigated in juvenile animals.

When using a macroscopic measure for wear, measuring the actual crown height (as opposed to measuring the height as done in the present study, as the distance to the bony alveolar margin) would have been a more insightful measure of crown tissue loss. Nevertheless, the shape of the height-age relationships of the present study (**Fig 18, S28, S29 Figs in S1 File**) corresponds well to plots given in Fig 13 of Laws [38] that relate tooth weight to age: during growth/eruption, there is an increase, followed by a decrease due to wear. To what degree the addition of cementum to the roots compensates for crown wear in hippos, as described in other mammals [55], is unclear. Laws [38] describes both, the addition of cementum layers and the resorption of dental roots with increasing age.

The observation that tooth row length (**Fig 17**) does not scale as expected from geometry, but with negative allometry within the common hippos (in which a sufficient number of animals was available to test this), corresponds to similar observations within other species [44]. Actually, hippos have been part of the discussion on whether dwarf species retain relatively larger teeth compared to their ancestral, larger-sized relatives [56, 57]. Whether pygmy hippos actually have larger relative cheek teeth than common hippos depends on whether skull and

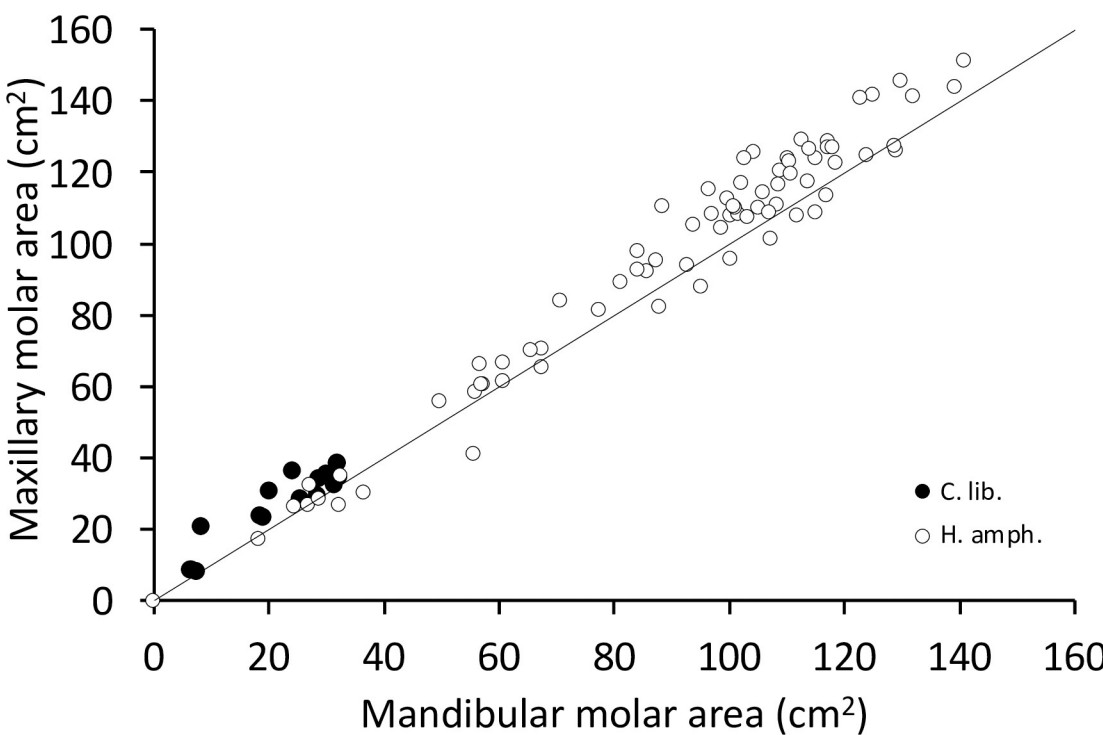

**Fig 16. Comparing the maxillary and mandibular molar tooth area in hippos (*Hippopotamus amphibius and Choeropsis liberiensis*).** The line depicts y = x. Note that the maxillary molar area is generally larger.

mandible dimensions actually relate geometrically to body mass in these species [57]–a question that cannot be resolved with our dataset as body masses of the museum specimens had generally not been recorded. Nevertheless, the fact that the scaling between the two species tended to be of a negative allometry for the tooth row length but a positive allometry for the 'diastema' length corresponds to dwarfing effects among related species. As reviewed in [44], a mismatch in tooth size and body size is the more likely the more recent the deviation of dwarfs or giants from the ancestral state in evolutionary time. The current interpretation of such findings is that tooth dimensions are more refractory to evolutionary changes in size than overall body size [44].

It has frequently been noted that hippo canines and incisors grow continuously [4, 5, 29, 46, 47]. In ever-growing teeth, growth is most likely regulated by pressure from the opposing tooth, so that wear and growth are balanced [reviewed in 58, 59]. Teeth of the anterior dentition that are worn down, by their own action or inadvertently by attrition caused by movement of the jaw for grinding chewing, are often rootless (i.e., ever-growing), to compensate for the continuous wear [11]. This mechanism appears intuitive for the canines, whose wear facets are in constant, opposing contact, and in which the wear facet represents the tip of the growing tooth. In hippos, it has been observed repeatedly that the loss of a canine leads to the uncontrolled growth of its antagonist, or that a mispositioned canine that does not reach contact with its antagonist, may grow excessively [29, 46, 49, 53, 60, 61].

However, for the ever-growing incisors, the regulatory mechanism does not appear as intuitive: the mandibular incisors of both species, as well as the maxillary incisors of common hippos, do not have a wear facet at their tip where a pressure signal by an antagonist is received. Rather, these teeth are in oblique or lateral contact with their antagonists, and it is unclear to what extent this lateral pressure signal will control tooth growth. The observations of

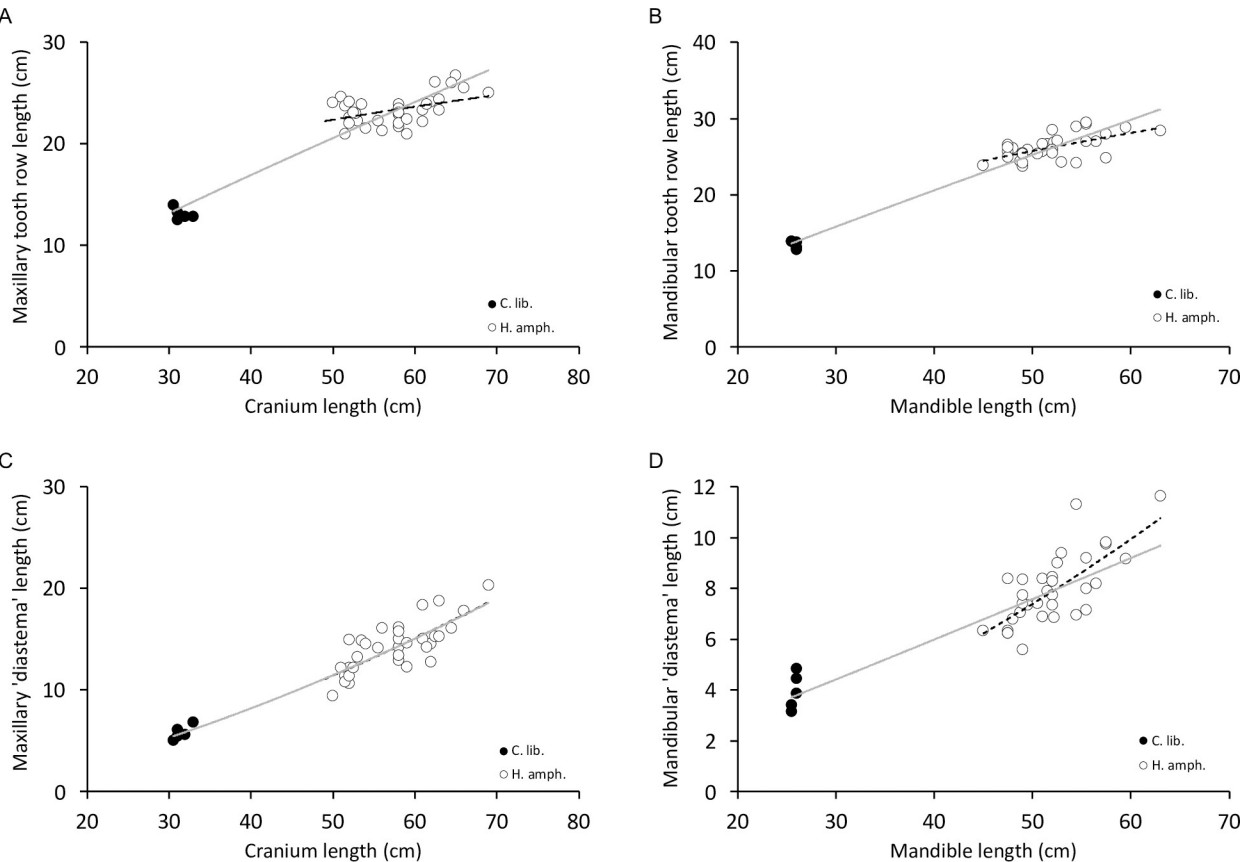

**Fig 17.** Scaling relationships of the (A) maxillary and (B) mandibular cheek tooth row (P2-M3) or the 'diastema' space (any non-tooth space between the P3 and the closest incisor) for the (C) cranium and (D) the mandible in adult common hippo (*Hippopotamus amphibius*) and pygmy hippo (*Choeropsis liberiensis*). The scaling exponents (with 95%CI) of the grey line (across all individuals of both species) are (A) 0.87 (0.76,0.98), (B) 0.92 (0.84,0.99), (C) 1.50 (1.34,1.66), (D) 1.06 (0.89,1.23); of the dotted line (within *H. amphibius*) (A) 0.31 (0.09,0.53), (B) 0.48 (0.25,0.71), (C) 1.52 (1.07,1.98), (D) 1.62 (1.07,2.17). If the 95%CI of the scaling exponent is < 1, this indicates negative allometry, if it is > 1, it indicates positive allometry.

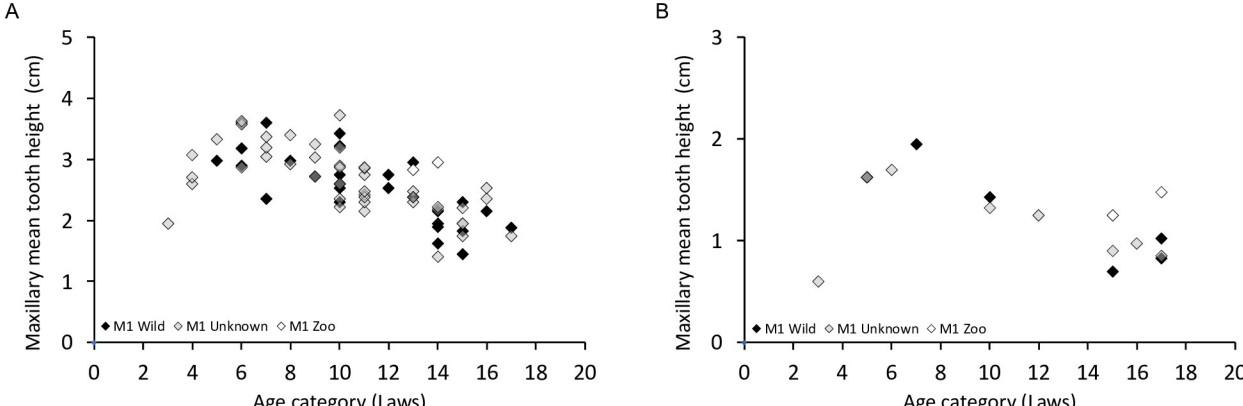

**Fig 18.** Examples of the relationship of tooth height and estimated age class for the maxillary M1 in (A) common hippo (*Hippopotamus amphibius*); (B) pygmy hippo (*Choeropsis liberiensis*). For the full set of teeth, see **S28, S29 Figs in S1 File**.

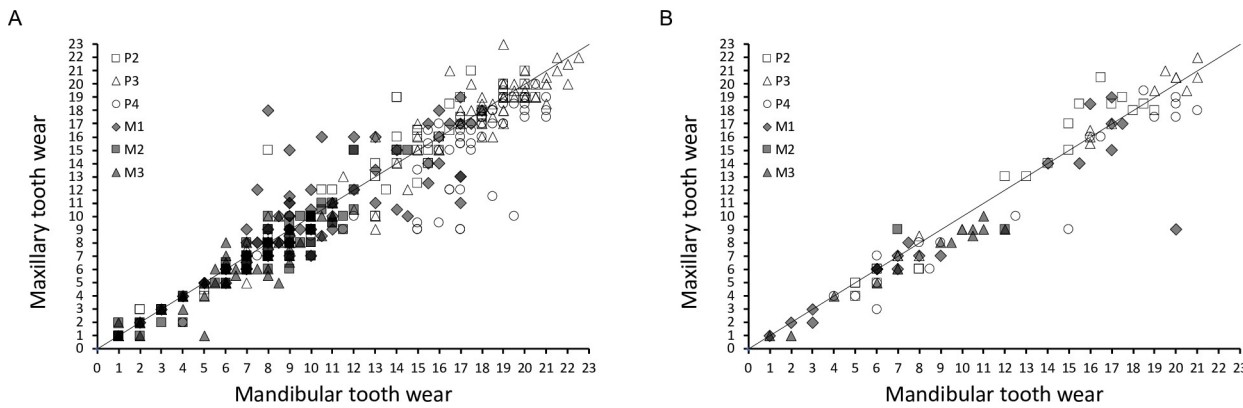

**Fig 19. Comparison of the maxillary and mandibular tooth wear grouped by individual teeth.** The line depicts y = x. Note no systematic difference in the wear of the respective teeth. (A) common hippo (*Hippopotamus amphibius*); (B) pygmy hippo (*Choeropsis liberiensis*).

mispositioned and overlong incisors, e.g. the I$^2$ in **Fig 4A**, a similar configuration of an overlong I$^2$ in [61], and the anecdotal observation that especially mandibular incisors are shortened in some zoo animals by zoo veterinarians (e.g. **Fig 1**), suggest that such a control occurs but can be lacking under certain circumstances. Detailed descriptions of the growth of the incisors, problems in zoo animals due to the incisors, and the effect of lateral pressure are lacking. We can only speculate that in incisors characterized by lateral or oblique facets, the tip will, under normal conditions, regularly break off or be worn down by using these teeth for fighting, digging [38], or manipulating hard objects. To our knowledge, no detailed observations on digging, or on object manipulation with incisors exist for hippos; therefore, fighting appears as the most likely cause. Observations in both free-ranging and zoo hippos on incisor use would be welcome.

The notion that the chewing of hippos is mainly orthal has, to our knowledge, first been suggested in 1878 [10]. This author also suggested that anisognathy–a difference in the width of the upper and lower jaw—is one of the necessary preconditions for a lateral chewing movement. For common hippos, this author reported no difference in this width (which was unfortunately not measured in the present study). Similarly, it can be gleaned from Fig 29 in

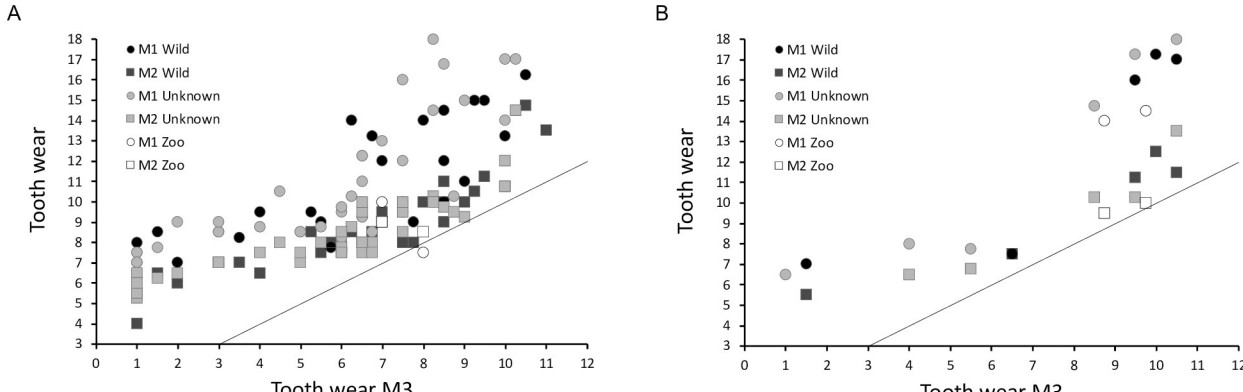

**Fig 20. Tooth wear of the first and second molar (means of maxillary and mandibular teeth per individual) in comparison to that of the third molar.** Data are separated by the known or unknown origin of the individuals. Black color = individual originated from the wild, white = individual was kept in a zoo, grey = origin of the individual is unknown. (A) common hippo (*Hippopotamus amphibius*); (B) pygmy hippo (*Choeropsis liberiensis*).

conjunction with Table 11 in Fortelius [42] that this author found hardly any difference between the jaw widths in common hippos. The only measure of the present study on the width of the jaws was taken at the base of the canines; this measure yielded opposing patterns, being wider for the lower jaw in common and for the upper jaw in pygmy hippos (**S22 Fig in S1 File**), in line with a slight lateral movement in the latter. Fortelius [42] argued that aniso-donty–a difference in the width between antagonists in the cheek tooth row–was another pre-condition for a lateral chewing movement, and reported a ratio of 1.32 for the common hippo–higher than our value of 1.19 for the common and 1.25 for the pygmy hippo. These values are comparatively low among ungulates; for example, clearly laterally chewing ruminants, equids or rhinoceroses have anisodonty ratios of 1.5–2.0 [42]. This supports the concept of little lateral chewing motion in hippos, and the difference between the two species again qualitatively reflects the observation that pygmy hippos show some limited degree of lateral jaw motion.

Other anatomical correlates for a restricted lateral chewing movement in hippos include the configuration of the mandibular joint: the mandibular ('glenoid') fossa of the common hippo is described as short and deep, indicating a joint that only allows orthal movement, whereas that of the pygmy hippo is described as long and shallow, facilitating more lateral movement [40]. Based on the correspondence of the configuration of this joint and the incisor wear patterns, Stuenes [40] groups fossil hippos as those with and without a lateral component in their chewing motion. We do not follow this author in describing the situation in the pygmy hippo as one of 'extensive' lateral movement, but prefer considering it a limited lateral component in a mainly orthal chewing pattern (when compared to other herbivores).

The lack of a major lateral chewing stroke could help explain several characteristics of hippo teeth that have been ascribed to other causes. Most particularly, when considering the incisor wear facets of hippos as an indication of a relic of a lateral motion, the resulting hypothesis that their orthal chewing movement is a secondary evolutionary feature derived from an ancestral condition of lateral chewing could help reconcile some conceived contradictions in understanding hippopotamid phylogeny and dental morphology.

Comparing the dental anatomy of the common hippo to other grazing animals, several authors noted that hippos have comparatively low-crowned teeth [42, 62–64]. The reasons for this have been sought in three characteristics so far by these authors: (i) The comparatively low food intake of hippos [30] should lead to overall less tooth wear. (ii) It has been suggested that hippo lawns are often immersed in water, which supposedly removes dust and grit–a hypothesis that gains support from the observation that hippo stomachs investigated by Langer [16, 65] did not contain sand (Langer, pers. comm.) although the stomach anatomy would predispose them for sand accumulation [66]; note however that Laws [38] specifically states that the feeding mode of the common hippo, with the plucking of grass by the lips, is prone to pull out roots and adhering soil, and that incisors are used for digging and common hippos ingest earth; evidently, more studies on soil ingestion by hippos would be welcome. (iii) Finally, it has been suggested that rather than evolving high tooth crowns hippos evolved the alternative adaptation of particularly thick enamel [12]. Without judging the value of these hypotheses, we suggest that an orthal chewing movement, with a reduced component of attrition (teeth gliding across teeth), is another factor contributing to the low degree of hypsodonty in hippos.

Several characteristics of the hippo dentition and skull have been described as secondary adaptations. A secondary simplification in the enamel schmelzmuster has been described towards radial enamel that is considered particularly susceptible to tensile stress [67]; tensile stress may occur less in the absence of lateral chewing movements. A secondary loss in enamel schmelzmuster complexity is considered a rare event in mammals [68]. Rather than being linked to an aquatic lifestyle or a change towards a grass diet–explanations that appear weak to

us given that other animals of the respective characteristic do not consistently show a reduction in enamel schmelzmuster complexity [67]–this secondary loss might be related to a secondary loss of the forces exerted by a lateral chewing stroke.

Many artiodactyls have molars that are called selenodont for the half moon-shaped pattern of their enamel ridges [42]. Such structures are typically associated with a lateral grinding chewing motion, where the two opposing selenodont occlusal surfaces glide across each other. Because this motion has a directionality (the direction of the lateral chewing stroke), these tooth surfaces are typically not bilaterally symmetrical in relation to a mesiodistal axis across the middle of the occlusal surface (separating the buccal paracone and metacone from the lingual protocone and hypocone [69]), but both parts of the tooth, the lingual as well as the buccal part, rather show a similar pattern that faces the same direction. The observation that hippopotamid ancestors had a selenodont occlusal surface, yet extant hippos have a trifoliate enamel surface pattern that actually is somewhat bilaterally symmetrical in relation to a mesiodistal axis (e.g. **S1; S6; S7; S30; S34-S37 Figs in S1 File**) [70–72], appeared difficult to reconcile with each other [51]. Nevertheless, Colbert [73] hypothesized that an evolution of the trifoliate pattern from bunoselenodont ancestors was more parsimonious than its evolution from a bunodont suid pattern. The detailed required changes in dental anatomy when evolving a trifoliate from a selenodont pattern have been intricately described by Lihoreau et al. [70]. A concomitant change in the chewing movement might provide a proximal cause for these changes in dental anatomy. If we assume that hippos secondarily lost a lateral chewing stroke, then the evolution of a bilaterally symmetrical trifoliate occlusal surface might become understandable from a mechanical point of view: without relevant directionality of movement, the occlusal surface might be more efficient if consisting of areas completely encircled by enamel ridges. When the mandibular molars meet the maxillary ones in central occlusion, one would expect food to be equally pressed towards the buccal and the lingual side of the cheek tooth row, explaining a bilaterally symmetrical tooth surface. If this hypothesis was correct, one would expect to find indications of lateral jaw movement, as indicated by striations on wear facets [74], in the more selenodont hippopotamid ancestors, and reduced to no such striations in species with trifoliate molars.

Additional support for the notion of a secondary loss of lateral chewing derives from Fortelius [42], who suggested that members of the Anthracotheriidae–the likely candidates for hippopotamid ancestry [70, 71]–have a higher anisodonty index (Table 11 of his study) and also a higher anisognathy index (Fig 29 of his study) than hippopotamids, which would support the concept of a secondary loss of the lateral chewing stroke in the latter. It would probably be worthwhile determining these indices for a variety of hippopotamid ancestors and fossil relatives for which cranium and mandible are conserved for a specimen, and put these findings into context with the occlusal surface pattern. Considering the masseter muscle architecture and insertion sites, Herring and Herring [75] suggested that hippos *'secondarily developed a requirement for wide gape'*, which also point towards secondary modifications of their chewing apparatus.

The evident resulting question is: what is the reason for a 'secondary requirement for wide gape' and the associated adaptive advantages that outweigh the loss of a lateral chewing stroke? Given the severe limitations of hippos due to their low chewing efficacy outlined in the introduction, these putative advantages must be substantial. According to Herring [76], the wide gape in hippos is nearly exclusively used in intraspecific agonistic encounters. Laws [38] reported that hippos use their canines and incisors mainly for fighting. Possibly, a rigid jaw that can be opened widely but that does not 'give' laterally in combat encounters is important in intraspecific fights. Herring [76] summarized observations on hippo behaviour, linking the strong selection for intraspecific combat to high population densities and the fact that hippo

habitats–along water bodies–are usually of high quality and do not deteriorate due to hippo grazing, but are actually maintained in a productive state [33]. Nevertheless, we follow the astonishment of Herring [76] that the selective pressures for successful intraspecific combat would be so high as to compromise the feeding apparatus and the overall digestive efficiency. It is tempting to suggest that the emphasis on intraspecific sexual competition in the cranium and dentition of hippos, while suitable for a semi-aquatic niche, prevented this clade from surviving in, or conquering, terrestrial habitats with competitors of a more efficient digestive physiology.

## Supporting information

**S1 File. Supplementary material (Avedik_HippoTeeth_Supplement) contains supplementary text on age dependence, cranium and mandible width, eruption sequences, and on deciduous teeth, S1 Table, and S1-S38 Figs.**
(PDF)

**S1 Data. Original data is provided as an excel file Avedik_HippoTeeth_RawData.**
(XLSX)

## Acknowledgments

We thank Loïc Costeur (Natural History Museum Basel), Martina Schenkel (Zoological Museum Zurich), Madeleine Geiger (the Natural History Museum St. Gallen), Christiane Funk (the Natural History Museum Berlin), Christian Montermann (the Zoological Research Museum Alexander Koenig Bonn), Irina Ruf and Katrin Krohmann (the Senckenberg Natural History Museum Frankfurt), Thomas Kaiser and Alexander Daasch (the Museum of Nature Hamburg), Manuela Schmidt and Bernd Bock (the Phyletisches Museum Jena), Albrecht Manegold (the State Museum of Natural History Karlsruhe), Stefan Merker and Carsten Leidenroth (the Stuttgart State Museum Natural History). We thank Brian Stefanski (Copenhagen Zoo) for common hippo videos, Christian Wenker and Bernhard Wörner (Basel Zoo) for support in filming the pygmy hippos, Michelle Aimée Oesch for photographs, and two reviewers for constructive comments.

## Author Contributions

**Conceptualization:** Annika Avedik, Marcus Clauss.

**Formal analysis:** Annika Avedik, Marcus Clauss.

**Investigation:** Annika Avedik.

**Methodology:** Marcus Clauss.

**Project administration:** Annika Avedik.

**Supervision:** Marcus Clauss.

**Visualization:** Annika Avedik.

**Writing – original draft:** Annika Avedik, Marcus Clauss.

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
