## [Decision Letter · Decision Letter 0]

2 Aug 2023

PONE-D-23-06724Chewing, dentition and tooth wear in Hippopotamidae (Hippopotamus amphibius and Choeropsis liberiensis)PLOS ONE

Dear Dr. Clauss,

Thank you for submitting your manuscript to PLOS ONE. After careful consideration, we feel that it has merit but does not fully meet PLOS ONE’s publication criteria as it currently stands. Therefore, we invite you to submit a revised version of the manuscript that addresses the points raised during the review process.

Reviewers agree on the quality of the submitted manuscript but spotted several minor points to be corrected. Please also consider improvements of the material and methods section according to reviewers comments. Please submit your revised manuscript by Sep 16 2023 11:59PM. If you will need more time than this to complete your revisions, please reply to this message or contact the journal office at plosone@plos.org. Please include the following items when submitting your revised manuscript:A rebuttal letter that responds to each point raised by the academic editor and reviewer(s). You should upload this letter as a separate file labeled 'Response to Reviewers'.A marked-up copy of your manuscript that highlights changes made to the original version. You should upload this as a separate file labeled 'Revised Manuscript with Track Changes'.An unmarked version of your revised paper without tracked changes. You should upload this as a separate file labeled 'Manuscript'.If applicable, we recommend that you deposit your laboratory protocols in protocols.io to enhance the reproducibility of your results. Protocols.io assigns your protocol its own identifier (DOI) so that it can be cited independently in the future. For instructions see: https://journals.plos.org/plosone/s/submission-guidelines#loc-laboratory-protocols. Additionally, PLOS ONE offers an option for publishing peer-reviewed Lab Protocol articles, which describe protocols hosted on protocols.io. Read more information on sharing protocols at https://plos.org/protocols?utm_medium=editorial-email&utm_source=authorletters&utm_campaign=protocols.

We look forward to receiving your revised manuscript.

Kind regards,

Cyril Charles

Academic Editor

PLOS ONE

Journal Requirements:

Reviewers' comments:

Reviewer's Responses to Questions

**Comments to the Author**

1. Is the manuscript technically sound, and do the data support the conclusions?

Reviewer #1: Yes

Reviewer #2: Partly

2. Has the statistical analysis been performed appropriately and rigorously? 

Reviewer #1: Yes

Reviewer #2: No

3. Have the authors made all data underlying the findings in their manuscript fully available?

Reviewer #1: Yes

Reviewer #2: No

4. Is the manuscript presented in an intelligible fashion and written in standard English?

Reviewer #1: Yes

Reviewer #2: Yes

5. Review Comments to the Author

Reviewer #1: This article is a thorough assessment of dental evidence relative to chewing movement in the two extant species of Hippopotamidae. This topic is of importance because it relates to dietary adaptations in a major group of megaherbivores characterized by a particular ecology, i.e. semiaquatic lifestyle, and with major bearing on African ecosystems – for the last five-seven years, many new papers explored the ecology of the common hippo, demonstrating its pervasive impact on continental water cycles and biodiversity. In addition, this study has implications for the evolutionary history of Hippopotamidae and of their anthracothere stem group, in which dietary changes (and notably grass consumption) co-occurred with increases of agonistic behavior. This paper provides further understanding of the anatomical trade-off between feeding and fighting that resulted in a successful expansion at the end the Miocene.

The paper is well-organized and well-written. Its illustrations are useful and clear. Above all, it includes a wealth of data. Notably, it provides for the first time a detailed assessment of anterior dentition occlusions and wear, and of their variations between individuals, hence going much further than the usual basic descriptions provided in literature. The study limits are well explained in the discussion, making the paper even more useful. Conclusions are sounds and congruent with results. This is overall a great contribution, and I believe it should be readily accepted. I just have spotted a few things that could be easily fixed as minor revisions.

Line 92: ref [32] is not discussing phylogenetic relationships of pygmy hippos with other hippos. This would be rather in ref [42].

Line 186: “Error! Reference source not found”

Please correct.

Fig. 13. For a more straightforward reading, each measurement position should be clearly indicated on this figure.

Regarding cheek tooth measurements, from text and supplementary pictures it is not totally clear how these were taken: including cingula or not (when present), maximal width, or at wear figures? It would be useful to describe the measurement protocol with the same clarity as for anterior dentition. Also, were highly worn cheek teeth measured or not? Length of M1s is often significantly altered through wear, was this considered?

Line 405: “the general impression is that the dP4 and the M1 have wider posterior cusps, and the M3 a wider anterior cusp”

This phrasing is a bit awkward: either this is true for all cases, or in most cases. For the latter it would be certainly easy to indicate the % of specimens this is not true.

Line 533. “We can only speculate that in incisors characterized by lateral or oblique facets, the

tip will, under normal conditions, regularly break off or be worn down by using these teeth

for digging and fighting, or manipulating hard objects”.

The authors should mentioned which hypothesis they favor here, based on literature and/or on their own observations. I have observed incisor breakages in several common hippos, and always assumed they were related to fight. Digging may indeed occur in Choeropsis liberiensis, but I am not sure we know how they proceed. Laws mentioned digging in Hippopotamus amphibius, but as far as I remember he is the only one and I did not read any account describing in details such digging. I do not recall any mentions of manipulating hard objects. If wear was caused by digging or hard object manipulating, it should have distinctive, recognizable patterns, and this could be tested.

Line 539 and following. The use of bicanine width compared in both species cannot be considered as a reliable inference of anisognathy, the latter being based on molars (e.g., Fortelius used M1s). Many hippopotamids differ in the relative expansion of the canine processes, and this projection can be much more readily seen as adaptive to agonistic behavior, little related to feeding, and thus largely uncorrelated with distance between molar rows. It would be more cautious to drop this point.

Lines 588-591. This is not totally clear. Dietary changes toward more grass eating have been actually recorded in conjunction with loss of schmelzmuster complexity. Co-occurrence of such dietary change with loss of lateral chewing movement may accompany lesser sensitivity to wear and greater susceptibility to tensile stress. These hypotheses are not mutually exclusive, but instead likely to reinforce each other.

Reviewer #2: In this interesting study, the authors discussed the chewing capacity of the two extant species of hippos based on the direct observations of living animals, on measurements on dentition and on description of dental wear on died specimens (i.e. skulls).

The introduction and discussion are well written (even if there are a few digressions at the end of the paper p.29-30), and we understand that the main question is the chewing capacity of hippos, rather than a focus on dental anatomy, which is only complementary.

This study follows a recent publication about the same topic on mammals, in which the same authors only briefly introduced the case of hippos. In that respect, the authors should more clearly state what part of data shown in the present paper was already published.

I am a bit confused by the "material and methods" section, which needs more details (e.g number of living animals filmed, what is direct observations, what is actually quantified). I am notably surprised that some measurements (e.g. cranial and mandible dimensions, gaps between premolars and between canines/incisors, height of cheek teeth) mentioned in the methods were not presented in the results (if we except supp. figures). If they are not relevant here (which seems to be the case), they should not be included in the paper. Moreover, all measurements shown in figures must be presented in supplementary tables, and statistical tests are also needed for accurate comparisons (e.g. upper vs lower dentition).

The authors also studied young to adult specimens based on dental wear stages, but we have no information regarding age classes for each specimen (based on Laws' table or on relative macrowear stages defined by the authors); it should be detailed accordingly. This information on the relative age of specimens could be of high interest to compare the chewing capacity of young and adult specimens according to the relative development of anterior teeth and snout.

I have also minor comments and suggestions which are listed in the pdf file.

Helder GOMES RODRIGUES

6. PLOS authors have the option to publish the peer review history of their article (what does this mean?). If published, this will include your full peer review and any attached files.

Reviewer #1: No

Reviewer #2: No

---

## [Author Response · Author response to Decision Letter 0]

16 Aug 2023

please see the attached reply letter

---

## [Decision Letter · Decision Letter 1]

1 Sep 2023

PONE-D-23-06724R1Chewing, dentition and tooth wear in Hippopotamidae (Hippopotamus amphibius and Choeropsis liberiensis)PLOS ONE

Dear Dr. Clauss,

Thank you for submitting your manuscript to PLOS ONE. After careful consideration, we feel that it has merit but does not fully meet PLOS ONE’s publication criteria as it currently stands. Therefore, we invite you to submit a revised version of the manuscript that addresses the points raised during the review process.

 Thank you also for your answers to the various comments. However, could you please consider the remaining at least the first comment of Reviewer 1, which I think needs to be addressed before publication ?

We look forward to receiving your revised manuscript.

Kind regards,

Cyril Charles

Academic Editor

PLOS ONE

Journal Requirements:

Reviewers' comments:

Reviewer's Responses to Questions

**Comments to the Author**

1. If the authors have adequately addressed your comments raised in a previous round of review and you feel that this manuscript is now acceptable for publication, you may indicate that here to bypass the “Comments to the Author” section, enter your conflict of interest statement in the “Confidential to Editor” section, and submit your "Accept" recommendation.

Reviewer #1: (No Response)

Reviewer #2: All comments have been addressed

2. Is the manuscript technically sound, and do the data support the conclusions?

Reviewer #1: Yes

Reviewer #2: Yes

3. Has the statistical analysis been performed appropriately and rigorously? 

Reviewer #1: Yes

Reviewer #2: Yes

4. Have the authors made all data underlying the findings in their manuscript fully available?

Reviewer #1: Yes

Reviewer #2: Yes

5. Is the manuscript presented in an intelligible fashion and written in standard English?

Reviewer #1: Yes

Reviewer #2: Yes

6. Review Comments to the Author

Reviewer #1: Thank you for addressing my comments. I would like to come back to two points which were in my opinion not fully addressed. This is minor, but will certainly not hurt your paper to be included.

My initial comment

Line 405: “the general impression is that the dP4 and the M1 have wider posterior cusps, and the M3 a wider anterior cusp”

This phrasing is a bit awkward: either this is true for all cases, or in most cases. For the latter it would be certainly easy to indicate the % of specimens this is not true.

Your response

Fair point. We refer to the corresponding Suppelementary Figures (S12-S16). We do not want to give the percentages, because this would mean that for each tooth position, we have to give the % for the mandibular and the maxillary one, for each species – this becomes a bit of confusing text for something that is of little value for the subsequent narration. If the editor wants, we can add this easily.

My rebuttal

This answer is not totally satisfying. First, there are no supplementary figure illustrating the proportions of the dP4. Second, a “general impression” cannot be trusted. Again, I would prefer to see some kind of quantification: “dP4 and M1 have usually wider posterior cusps (in XX % and XX % of the cases, respectively), and the M3 a wider anterior cusp (in XX % of the cases).” This is not confusing, and this data could be useful for readers. Please add.

My initial comment

Lines 588-591. This is not totally clear. Dietary changes toward more grass eating have been actually recorded in conjunction with loss of schmelzmuster complexity. Co-occurrence of such dietary change with loss of lateral chewing movement may accompany lesser sensitivity to wear and greater susceptibility to tensile stress. These hypotheses are not mutually exclusive, but instead likely to reinforce each other.

Your response

Given that this is hypothetical, we would like to not use this argument by the reviewer.

1. We think that a change to grass AND a loss of lateral chewing movement is not something that occurred often, but rather quite rarely (because most large herbivores use some kind of transversal chewing movement).

2. We cite work that actually no rule of enamel schmelmuster complexity and diet seems to apply.

3. If, as the reviewer states, the move towards an orthal chewing movement should increase tensile stress (as opposed to our hypothesis that it reduces it, which we find more compelling because simple ‘pressur’ without a lateral compenonent does not sound like causing a lot of tensile stress), then how would this match the observation that the schmelzmuster of hippos appears particularly susceptible to tensile stress?

My rebuttal

I do not think that the authors understood what I meant, and I am sorry if my phrasing was not clear enough. Let me explain differently.

According to the reference cited by the authors [now 67], loss of schmelzmuster has been document twice in hippopotamoids, in both cases contemporaneously with a dietary change. These authors therefore suggested that secondary radial enamel was an adaptation to dietary changes (from a less to a more abrasive diet, i.e. including more grass) in place of the usual acquisition of hypsodonty seen in other ungulates; indeed hippopotamoids did not developed markedly hypsodont molars.

Loss of schmelzmuster induces a greater sensitivity to tensile stress. What I meant initially is that the loss of lateral movement (a more orthal chewing movement) could be help mitigate this greater sensitivity, in accordance to the authors’ hypothesis (and not in disagreement with it, contra their reading of my comment). This loss of lateral movement could have accompanied other forms of mitigation (crystallite decussation).

I fully agree that this is hypothetical, but a discussion is the right place to discuss hypotheses as long as they are testable, and it is evident that the authors did exactly this in the first place. Unlike the co-occurrence of schmelzmuster losses and dietary changes in two different lineages of hippopotamoids, we do not know yet if losses of lateral movement occurred at the same time, but I am convinced it is worth mentioning this possibility here, because it can be tested.

Reviewer #2: My main concerns regarding the material used, the different measurements detailed and the consideration of both young and adult specimens in relation to chewing capacities have been addressed by the authors. I would like to acknowledge them for that effort.

I would like to precise that I just wanted to warn the authors that they should be careful when applying the scheme of the common hippo to the pygmy hippo for class age estimates (cf Laws, 1968), because their dental eruption pattern slightly differ (cf eruption of M2) and could also influence dental wear and thus their relative age. I thought the present data would permit to overcome this potential issue, using both dental eruption and wear to define relative class age estimates for the pygmy hippo. By the way, it was not mandatory to mention dental eruption patterns in supplementary data. Sorry, it was not clear enough.

Minor point:

l.583 Write “Nevertheless”

7. PLOS authors have the option to publish the peer review history of their article (what does this mean?). If published, this will include your full peer review and any attached files.

Reviewer #1: No

Reviewer #2: No

---

## [Editor Report · Decision Letter 2]

6 Sep 2023

Chewing, dentition and tooth wear in Hippopotamidae (Hippopotamus amphibius and Choeropsis liberiensis)

PONE-D-23-06724R2

Dear Dr. Clauss,

We’re pleased to inform you that your manuscript has been judged scientifically suitable for publication and will be formally accepted for publication once it meets all outstanding technical requirements.

Kind regards,

Cyril Charles

Academic Editor

PLOS ONE
---

## [Editor Report · Acceptance letter]

13 Sep 2023

PONE-D-23-06724R2 

Chewing, dentition and tooth wear in Hippopotamidae (*Hippopotamus amphibius and Choeropsis liberiensis*) 

Dear Dr. Clauss:

I'm pleased to inform you that your manuscript has been deemed suitable for publication in PLOS ONE. Congratulations! Your manuscript is now with our production department. 

Kind regards, 

on behalf of

Dr. Cyril Charles 

Academic Editor

PLOS ONE